



# LPJmL4 - a dynamic global vegetation model with managed land: Part II – Model evaluation

Sibyll Schaphoff[1], Matthias Forkel[2], Christoph Müller[1], Jürgen Knauer[3], Werner von Bloh[1], Dieter Gerten[1,4], Jonas Jägermeyr[1], Wolfgang Lucht[1,4], Anja Rammig[5], Kirsten Thonicke[1], and Katharina Waha[1,6]

[1]Potsdam Institute for Climate Impact Research, Telegraphenberg, PO Box 60 12 03, 14412 Potsdam, Germany
[2]TU Wien, Climate and Environmental Remote Sensing Group, Department of Geodesy and Geoinformation, Gusshausstraße 25-29, 1040 Wien, Austria
[3]Max Planck Institute for Biogeochemistry, Hans-Knöll-Str. 10, 07745 Jena, Germany
[4]Humboldt Universität zu Berlin, Department of Geography, Unter den Linden 6, 10099 Berlin, Germany
[5]Technical University of Munich, Germany
[6]CSIRO Agriculture & Food, 306 Carmody Rd, St Lucia QLD 4067, Australia

*Correspondence to:* Sibyll.Schaphoff@pik-potsdam.de

**Abstract.** The dynamic global vegetation model LPJmL4 is a process-based model that simulates climate and land-use change impacts on the terrestrial biosphere, the water and carbon cycle and on agricultural production. Different versions of the model have been developed and applied to evaluate the role of natural and managed ecosystems in the Earth system and potential impacts of global

environmental change. A comprehensive model description of the new model version, LPJmL4, is provided in a companion paper (Schaphoff et al., submitted). Here, we provide a full picture of the model performance, going beyond standard benchmark procedures, give hints of the strengths and shortcomings of the model to identify the need of further model improvement. Specifically, we evaluate LPJmL4 against various datasets from in-situ measurement sites, satellite observations,

and agricultural yield statistics. We apply a range of metrics to evaluate the quality of the model to simulate stocks and flows of carbon and water in natural and managed ecosystems at different temporal and spatial scales. We show that an advanced phenology scheme improves the simulation of seasonal fluctuations in the atmospheric $CO_2$ concentration while the permafrost scheme improves estimates of carbon stocks. The full LPJmL4 code including the new developments will be supplied

Open Source through a Gitlab repository. We hope that this will lead to new model developments and applications that improve model performance and possibly build up a new understanding of the terrestrial biosphere.



## 1 Introduction

The terrestrial biosphere is a central element in the Earth System, supporting ecosystem functioning
and also providing food to human societies. Dynamic global vegetation models (DGVMs) have been
developed and used to study the biosphere dynamics under climate and land-use change. LPJmL4
is a DGVM with managed land that has been developed to investigate potential impacts of climate
change on the terrestrial biosphere including natural and managed ecosystems, and is now described
in full detail in the companion paper (Schaphoff et al., submitted). LPJmL and its predecessors
have been originally benchmarked against ecosystem carbon and water fluxes and global maps of
vegetation distribution (Sitch et al., 2003), against runoff (Gerten et al., 2004), agricultural yield
statistics (Bondeau et al., 2007), satellite observations of fire activity (Thonicke et al., 2001, 2010),
permafrost distribution and active layer thickness (Schaphoff et al., 2013), satellite observations of
FAPAR and albedo (Forkel et al., 2014, 2015), and atmospheric $CO_2$ concentrations (Forkel et al.,
2016). These previous evaluation studies focussed on single processes or components of the model.
Here we present now a comprehensive multi-sectoral evaluation to demonstrate that LPJmL4 can
consistently represent multiple aspects of biosphere dynamics.

LPJmL4 spans a wide range of processes (ranging from biogeochemical to ecological aspects,
from leaf-level photosynthesis to biome composition) and combines natural ecosystems, terrestrial
water cycling, and managed ecosystems in one consistent framework. As such, it is increasingly
applied for cross-sectoral studies such as the quantification of planetary boundaries (Steffen et al.,
2015) and of multidimensional impacts of climate and land use change (e.g., Gerten et al., 2013;
Ostberg et al., 2015; Warszawski et al., 2014; Zscheischler et al., 2014; Müller et al., 2016). With
this complexity, its evaluation against historical observations along multiple dimensions is essen-
tial (Harrison et al., 2016). For such purpose, standardized benchmarking systems have been pro-
posed (Luo et al., 2012; Kelley et al., 2013; Abramowitz, 2005). In the present evaluation of a broad
range of fundamental features of the LPJmL4 model, we basically follow the benchmarking proce-
dures, variables, performance metrics and diagnostic plots suggested by Luo et al. (2012), and Kelley
et al. (2013), respectively. Thus the presented evaluation is going well beyond earlier evaluations of
DGVMs and of LPJmL (and its predecessors) itself. We pay special attention to LPJmL4's capability
to reproduce observed seasonal and interannual dynamics and patterns of key biogeochemical, hy-
drological and agricultural processes at various spatial scales. In so doing, we highlight the model's
unique feature of representing the interaction of processes for both natural and agricultural ecosys-
tems in a single, internally consistent framework.

## 2 Model benchmark

In the following we describe in detail the model benchmarking scheme employed here, which allows
for a consistent evaluation of processes simulated by LPJmL4 at seasonal and annual resolution





and at spatial scales from site level (using e.g. eddy-flux measurements for comparison) to global level (using e.g. remote sensing products). The evaluation spans the time period from 1901 to 2011.

The benchmarking analysis also considers results from different model set-ups and previous model versions, in order to demonstrate advancements achieved with the current LPJmL4 version and the sensitivity of results to individual new modules.

### 2.1    Model setup and simulation experiments

As described in Schaphoff et al. (submitted), we drive the model simulations with observation based

monthly input data on daily mean temperatures from Climatic Research Unit (CRU TS version 3.23 University of East Anglia Climatic Research Unit; Harris (2015); Harris et al. (2014)), precipitation provided by the Global Precipitation Climatology Centre (GPCC Full Data Reanalysis Version 7.0, (Becker et al., 2013)). Shortwave downward radiation and net downward longwave radiation are reanalysis data from ERA-Interim (Dee et al., 2011). Monthly average wind speeds are based on

NCEP re-analysis data and were regridded to CRU (NOAA-CIRES Climate Diagnostics Center, Boulder, Colorado, USA, Kalnay et al. (1996b)). The number of wet days per month, which is used to allocate monthly precipitation data to individual days of the corresponding months, is derived synthetically as suggested by New et al. (2000). Dew point temperature is approximated from daily minimum temperature (Thonicke et al., 2010).

The spatial resolution of all input data is 0.5° and the model simulations are conducted at this spatial resolution. All model simulations are based on a 5000 year spinup simulations after initializing all pools to zero. A second spinup simulation of 390 years is conducted in which human land use is introduced in 1700, using the data of Fader et al. (2010). In addition to the original data set description of Fader et al. (2010), sugar cane is now represented explicitly. Soil texture is given by

the Harmonized World Soil Database (HWSD) version 1 (Nachtergaele et al., 2008) and parameterized based on the relationships between texture and hydraulic properties from Cosby et al. (1984). The river routing scheme is from the simulated Topological Network (STN-30) drainage direction map (Vorosmarty and Fekete, 2011). Reservoir parameters are taken from Biemans et al. (2011), locations are obtained from the GRanD database (Lehner et al., 2011).

We test the influence of specific processes that have been implemented or improved to contribute to the new developments in LPJmL4 (specifically, permafrost, phenology, and fire) on overall model performance by conducting the following factorial experiments:

– LPJmL4-GSI-GlobFIRM: a simulation with all standard model features enabled as used in Schaphoff et al. (submitted), i.e. with land use, permafrost dynamics, the growing season index

(GSI) phenology scheme and the simplified fire model (GlobFIRM). This model experiment is the default LPJmL4 model experiment.





- LPJmL4-GSI-GlobFIRE-PNV: same, but for potential natural vegetation (PNV) to evaluate the role of managed land on global pattern and processes. This model experiments mimics the original LPJ model (i.e. without agriculture) but with improved phenology.

- LPJmL4-NOGSI-GlobFIRM: a simulation with land use, permafrost dynamics and the simplified fire model, but without the GSI phenology for testing the sole effect of the GSI phenology. Instead of the GSI phenology, here we use the original phenology model (Sitch et al., 2003) that is based on a growing-degree day approach. This experiment mimics the LPJmL 3.5 version (including the LPJ core, agriculture, and permafrost) as described in Schaphoff et al.
(2013).

- LPJmL4-NOGSI-NOPERM-GlobFIRM: a simulation with land use and the simplified fire model but without permafrost and without the GSI phenology. This model experiment mimics the original LPJmL 3.0 model with the LPJ core (Sitch et al., 2003) and the agricultural modules (Bondeau et al., 2007).

- LPJmL4-GSI-SPITFIRE: a simulation setup as LPJmL4-GSI-GlobFIRM but with the process-based fire model (SPITFIRE). This experiment is a LPJmL4 model run with an alternative fire module.

## 2.2 Evaluation data sets

Following Kelley et al. (2013) we compare LPJmL4 simulations against independent data for vege-
tation cover, atmospheric $CO_2$ concentrations, carbon stocks and fluxes, fractional burnt area, river discharge and fraction of absorbed photosynthetically active radiation (FAPAR). Beyond these suggestions of Kelley et al. (2013), we extend the benchmarking system to data sets of eddy flux tower measurements of evapotranspiration and net ecosystem exchange rate (NEE). Ecosystem respiration $Re$ is evaluated against both eddy-flux measurements and operational remote sensing data. Crop
yields are evaluated against FAOstat data (FAO, 2014). For FAPAR, we use not just one but three different reference data sets to account for uncertainties from multiple satellite datasets (see Section 2.2.6). We also compare LPJmL4 results against data that are not fully independent of other models (mostly empirical, data-driven modelling concepts), acknowledging the limitations of these data in a benchmark system. However, this allows for assessing LPJmL4's performance in additional as-
pects, where fully data-based products are not available. These data comprise global gridded data sets of vegetation or aboveground biomass carbon (Carvalhais et al., 2014; Liu et al., 2015), cropping calendars (Portmann et al., 2010), global GPP (Jung et al., 2011), $Re$ (Jägermeyr et al., 2014), soil carbon (Carvalhais et al., 2014), and evapotranspiration (Jung et al., 2011).

We use both site-level and global gridded data because they provide complementary information
but have different advantages for the comparison with simulated data like from LPJmL4. Site-level data are fully independent from model estimates and assumptions, but typically only represent a spe-





cific ecosystem with a certain vegetation and soil type, and a own site history. Thus site-level data
has only a limited representativeness for 0.5° grid cells. On the other hand, global gridded data are
available at the same scale and thus can be directly compared to simulation outputs. However, global
gridded datasets usually rely on empirical modelling approaches and ancillary data to upscale and
extrapolate site-level data to large regions. Nevertheless specific site conditions like forest manage-
ment affecting site age, biomass, and carbon fluxes can be hardly re-simulated for a large number of
global sites within a DGVM. On the other hand global gridded products on GPP (Beer et al., 2010;
Jung et al., 2011) or $Re$ (Jägermeyr et al., 2014) provide information at the global application scale
of DGVMs. Although Kelley et al. (2013) reject the use of such datasets for model benchmarking
because they depend on modelling approach, we accept the additional use of such datasets because
they prevent the scale mismatch between site-level data and global DGVM simulations.

### 2.2.1 Vegetation cover

We compare simulated vegetation cover to the ISLSCP II vegetation continuous fields of Defries and
Hansen (2009) as suggested by Kelley et al. (2013). This data set is a gridded snapshot of vegetation
cover for the years 1992/1993 from remote sensing data and distinguishes bare soil, herbaceous, and
tree cover fractions. Tree cover fractions are further distinguished into evergreen vs. deciduous and
into broad-leaved vs. needle-leaved tree types, respectively. The herbaceous vegetation class includes
woody vegetation that is less than 5m tall. Data uncertainties increase, where tree cover is <20% due
to understorey vegetation and soil disturbing the signal, and above 80% due to signal saturation
(Defries and Hansen, 2009; Kelley et al., 2013). The data set was aggregated to 0.5° resolution
(Defries and Hansen, 2009; Kelley et al., 2013). To test if the simulated land cover of LPJmL4
performs better than a random-generated land cover distribution we compare the performance of
LPJmL4 also to the random model as suggested by Kelley et al. (2013, Section 2.3.5), whereas the
original dataset ISLSCP II vegetation continuous fields were randomly resampled.

### 2.2.2 Atmospheric CO$_2$ concentration

To evaluate the model's capacity to capture global-scale, intra- and interannual fluctuations of at-
mospheric CO$_2$ concentrations as driven by the uptake activity of the terrestrial biosphere, we com-
pare simulated CO$_2$ concentrations recorded at two remote continuous measurements at Mauna Loa
(MLO, 19.53°N, 155.58°W) and Point Barrow (BRW, 71.32°N, 156.60°W) (see Rödenbeck (2005)
for further details on these measurements). We use monthly CO$_2$ concentrations from flask and
continuous measurements from 1980 to 2010 for the comparison with LPJmL4 simulations. CO$_2$
observations were temporally smoothed and interpolated using a standard method (Thoning et al.,
1989). The atmospheric transport model (TM3, Rödenbeck et al. (2003)) in Jacobian representation
(Kaminski et al., 1999) simulates the global CO$_2$ transport using estimates of NBP (here simulated
by LPJmL4, see Forkel et al. (2016)), estimated net ocean CO$_2$ fluxes from the Global Carbon



Project (Le Quéré et al., 2015) and fossil fuel emissions from the Carbon Dioxide Information Analysis Center (CDIAC; Boden et al. (2013)). Atmospheric transport in TM3 is driven by wind fields of the NCEP reanalysis (Kalnay et al., 1996a) at a spatial resolution of $4° \times 5°$.

### 2.2.3 Terrestrial carbon stocks and fluxes

Model-independent reference data for carbon stocks and fluxes are available from Luyssaert et al. (2007) for various sites globally distributed. This data set comprises vegetation carbon, aboveground biomass, gross primary production (GPP) and net primary production (NPP). GPP flux data from Luyssaert et al. (2007) are based on eddy-flux measurements and are subject to those uncertainties, reported in Luyssaert et al. (2007, Table 2). Contrastingly, NPP data are derived from direct measurements of continuous leaf-litter collection, allometry-based estimates of stem and branch NPP from basal measurements, root NPP estimates from soil cores, mini rhizotrons, or soil respiration, and destructive understorey harvest. Estimates here are subject to uncertainties, depending on the sampling methods (Luyssaert et al., 2007). Several individual sites of this data set can be located within one simulation unit of a $0.5°$ grid cell and we thus compare simulated values to the range of site measurements in that grid cell.

Alternatively to the site-based GPP data from Luyssaert et al. (2007), we also compare spatial patterns and grid cell specific GPP simulations to the GPP data set of Jung et al. (2011), as also suggested by Kelley et al. (2013). This global data set is based on a larger set of eddy flux tower measurements than the data set of Luyssaert et al. (2007), but uses additional satellite and climate data, and empricial modelling for extrapolation to full global coverage. $Re$ is evaluated for the time period 2000 to 2009 directly against plot-scale FLUXNET measurements (ORNL DAAC, 2011), but also against large-scale $Re$ estimates from an empirical model based on operational remote sensing data by the Moderate Resolution Imaging Spectroradiometer (MODIS) with a resolution of 1 km and 8 days (Jägermeyr et al., 2014).

In addition to GPP, $R_e$ and NPP, we also compare simulated net ecosystem exchange (NEE) fluxes with eddy flux tower measurements directly. We use 70 time series of estimated NEE from eddy flux tower sites that measure the exchanges of carbon and water fluxes continuously over a broad range of climate and biome types (ORNL DAAC, 2011). Nevertheless, eddy flux tower sites are not well distributed across the globe and sites in the temperate and boreal zone are better represented than the tropical zone.

For the global comparison of the soil and vegetation carbon stocks we use the data compiled by Carvalhais et al. (2014). The soil organic carbon estimations are based on the Harmonized World Soil Database (HWSD) (Nachtergaele et al., 2008). Carvalhais et al. (2014) used a empirical model to calculate soil organic carbon stocks ($\mathrm{kg\,m^{-2}}$) from soil organic content (%), layer thickness (m, here for the first 3 m), gravel content (vol%), and bulk density ($\mathrm{kg\,m^{-3}}$). They pointed out that regions as North America and northern Eurasia are less reliable as HWSD was work in progress




at that time. The vegetation carbon data of Carvalhais et al. (2014) are based on a forest biomass
map for temperate and boreal forests from microwave satellite observations (Thurner et al., 2014), a

biomass maps for tropical forests based on Lidar observations (Saatchi et al., 2011), and an additional
estimate of grassland biomass. Uncertainties in biomass are in most regions between 30-40 % and
are strongly related to uncertainties in belowground biomass. We also compare simulated above-
ground biomass to the estimates of Liu et al. (2015), which is also based on satellite-based passive
microwave data. This comparison requires additional assumptions on the separation of above ground

and below ground biomass in LPJmL4 simulations. Liu et al. (2015) estimates for 2000 a global
above ground biomass at 362 PgC with a a 90 % confidence interval of 310–422 PgC.

### 2.2.4 Terrestrial water fluxes

River discharge measurements are taken from the ArcticNET and UNH/GRDC data sets for 287
gauges (Vörösmarty et al., 1996). From this data base, we only selected river gauges with catchment

areas $\geq 10{,}000\,\mathrm{km}^2$ as the model setup and resolution are not suitable for comparison with smaller
catchments. We also only selected river gauge records with a temporal coverage of more than 95 %
of the observation period and an observation period longer than 2 years at a monthly resolution.

    Evapotranspiration fluxes are taken from the FLUXNET data base and comprise 126 sites, of
which we selected sites (n=99) for which at least 3 years of recorded data are available. Additional

to site-level data, we used global gridded ET data from Jung et al. (2011), which is based on an
upscaling of site-level eddy covariance observations with satellite and climate data using a machine
learning approach.

    Irrigation withdrawal and consumption data are from other modelling approaches. Nonetheless,
human water use for irrigation is an important component in the terrestrial water cycle and we

discuss modelled LPJmL4 estimates in comparison to other model-based estimates, acknowledging
the limitation of this comparison and addressing different sources of uncertainty.

### 2.2.5 Permafrost

For the evaluation of simulated permafrost dynamics, we use the measured thaw depth data from 131
stations of the CALM station data set (Brown et al., 2000) as well as the IPA Circum–Arctic Map

of Permafrost (Brown et al., 1998). The distribution of permafrost is based on regional elevation,
physiography and surface geology. The permafrost extent represents four classes which categorize
the percentage of the ground underlain by permafrost (continuous, 90-100 %; discontinuous, 50-
90 %; sporadic, 10-50 %; and isolated patches of permafrost, 0-10 %).

### 2.2.6 Fractional area burnt

For the evaluation of simulated fire dynamics, we employ data on fractional area burnt from the
GFED4 data set (Giglio et al., 2013) for the period 1995 to 2014 and CCI Fire Version 4.1 (Chuvieco





et al., 2016) for the period 2005 to 2011. Mean annual burned area was computed for both datasets for the overlapping period (2005-2011). Both data sets are derived from satellite data. Active fire data was used in GFED4, to prolong the dataset prior to the MODIS period (i.e. for 1995-2000).

### 2.2.7   Fraction of absorbed photosynthetic active radiation and albedo

Data on the fraction of absorbed photosnthetically active radiation (FAPAR) are derived from three different satellite data sets to account for differences between datasets for model evaluation (see Table 4, Forkel et al. (2015)). The MODIS (Moderate-Resolution Imaging Spectroradiometer; USGS, 2001) FAPAR (Knyazikhin et al., 1999), the Geoland2 BioPar (GEOV1) FAPAR data set (Baret
et al., 2013) (hereafter called VGT2 FAPAR), and the GIMMS3g FAPAR data set (Zhu et al., 2013). The MODIS FAPAR data set is taken from the MOD15A2 product with a temporal resolution of 8 days at a spatial resolution of 1 km, covering the period 2001 to 2011. VGT2 is based on SPOT VGT with a temporal resolution of 10 days and 0.05° spatial resolution (Baret et al., 2013), covering the period 2003 to 2011. The GIMMS3g data set has a 15-day temporal resolution and 1/12° spatial
resolution and covers the period from 1982 to 2011. Data on FAPAR is also subject to uncertainties from the processing of the remotely sensed data and is not available continuously for all areas. We compare the spatial patterns of the peak FAPAR, and the temporal dynamics of FAPAR in each grid cell, and seasonal variations in FAPAR averaged for Köppen-Geiger climate zones for the three different FAPAR data sets. The aggregated FAPAR represents the average time series for all grid
cells that belong to a certain Köppen-Geiger climate zone (see also Forkel et al. (2015)). For the Köppen-Geiger climate zones, FAPAR time series are averaged over all grid cells that belong to that Köppen-Geiger climate zone (see also Forkel et al. (2015)).

For the evaluation of the reflectance of the earth surface we used the MODIS C5 albedo time series data set from 2000-2010 (Lucht et al., 2000; Schaaf et al., 2002), that we also aggregated to
Köppen-Geiger climate zones for the evaluation here.

### 2.2.8   Agricultural productivity

Detailed data on crop growth and productivity are available for individual sentinel sites (Rosenzweig et al., 2014). For global-scale or regional simulations, reference data are available only for crop yields and in (sub-)national aggregations (e.g., FAO, 2014) or as processed and interpolated
gridded products (Iizumi et al., 2014). In all yield data statistics outside of well-controlled field experiments, yield levels and interannual variability are not only affected by variability in weather, but also by variance in management conditions, such as sowing dates, variety choices, cropping areas, fertilizer inputs, pest control and others (Schauberger et al., 2016). Consequently, it is difficult to evaluate model performance from a comparison of simulated yields with static assumptions on most
management aspects with yield statistics in which the contribution of weather variability on yield variability is unknown. Müller et al. (2017) propose a combination of global gridded crop model





simulations and different observation-based yield data sets to establish a benchmark for global crop model evaluation. Generally, global gridded crop models perform well in most regions for which statistical models can detect significant influence of weather on crop yield variability (Ray et al.,

2015). We here evaluate LPJmL4 by comparing simulated and observed yield variability of the 10 top-producing countries (FAO, 2014). We refrain from comparing to individual sentinel sites, but refer to the evaluation of LPJmL crop simulations at global, national and grid cell scale in the global gridded crop model evaluation framework (Müller et al., 2017). As in (Müller et al., 2017), we here aggregate simulated grid-cell level yield time series to average national yield time series using the

MIRCA2000 data set for spatial aggregation (Porwollik et al., 2016) and removing trends in observations and simulations with a moving window average (see Müller et al. (2017) for details).

The productivity of biomass plantations is evaluated with data from experimental sites for miscanthus, switchgrass, poplar, willow and eucalyptus production, using the data collection of Heck et al. (2016). Data on biomass productivity typically report a data range. These are site-specific

management differences and reflect the diverse drivers of reported productivity, such as variation of plant species, fertiliser use and irrigation management, crop spacing or sapling size. We average the minimum and maximum values to derive the mean productivity per site.

### 2.2.9 Sowing dates

For evaluating the accuracy of the simulated rainfed sowing dates, we use the global data set of

growing areas and growing periods, MIRCA2000 (Portmann et al., 2008, 2010) at a spatial resolution of 0.5° and a temporal resolution of one month, as proposed by Waha et al. (2012). Monthly data in MIRCA2000 were converted to daily data by assuming that the growing period starts at the first day of the month following Portmann et al. (2010). MIRCA2000 reports several growing periods in a year for some administrative units and wheat, rapeseed, rice, cassava and maize. For comparison

we select the best corresponding growing period so that a close agreement indicates that simulated sowing dates are reasonable, but not necessarily the most frequently chosen by farmers. We do not compare simulated sowing dates for sugar cane (see SI-Fig.19) to observed sowing dates as MIRCA2000 assumes it is grown all year round as a perennial crop.

### 2.3 Evaluation metrics

We employ Taylor diagrams Taylor (2001) to compare the correlation, differences in standard deviation, and the centered root mean squared error (CRMS) between simulated and observed carbon and water fluxes at FLUXNET sites (ORNL DAAC, 2011) and at gauge stations from ArcticNET and UNH/GRDC. The standard deviations of the reference data sets have been normalized to 1.0 so that multiple sites can be displayed in one figure.

For global gridded reference data sets, such as for carbon stocks, we show spatial patterns in maps and aggregations as latitudinal means and quantify overall differences as a spatial correlation





**Table 1.** Evaluation metrics

| Metric | Equation | Reference |
|--------|----------|-----------|
| NMSE | $\mathrm{NMSE} = \frac{\sum_{i=1}^{N}(y_i - x_i)^2}{\sum_{i=1}^{N}(x_i - \overline{x})^2}$ | Kelley et al. (2013) |
| NME | $\mathrm{NME} = \frac{\sum_{i=1}^{N}|y_i - x_i|}{\sum_{i=1}^{N}|x_i - \overline{x}|}$ | Kelley et al. (2013) |
| ME | $\mathrm{ME} = \frac{\sum_{i=1}^{N}|y_i - x_i| \cdot A_i}{\sum_{i=1}^{N} A_i}$ | |
| W | $\mathrm{W} = 1 - \frac{\sum_{i=1}^{N}(y_i - x_i)^2 \cdot A_i}{\sum_{i=1}^{N}(|y_i - \overline{x}| + |x_i - \overline{x}|)^2 \cdot A_i}$ | Willmott (1982) |
| MM | $\mathrm{MM} = \frac{\sum_{i=1}^{N}|q_{i,j} - p_{i,j}|}{N}$ | Kelley et al. (2013) |

$y_i$ is the simulated and $x_i$ the observed value in grid cell $i$, $\overline{x}$ the mean observed value, $A_i$ the area weight in grid cell $i$, and $N$ the number of grid cells or sites, $q_{i,j}$ is the simulated and $p_{i,j}$ is the observed fraction of item $j$ in grid cell $i$. Normalized mean square error – NMSE, Normalized mean error – NME, ME – Mean absolute error, W – Willmott coefficient of agreement, MM – Manhattan metric

analysis over all grid cells (see Table 4). As suggested by Kelley et al. (2013) we use the normalized mean squared error (NMSE) to describe differences between model simulation and reference data sets. The NMSE is zero for perfect agreement, 1.0 if the model is as good as using the data mean

as predictor and larger 1.0 if the model performs less well than that. The squared error term puts stronger emphasis on large deviations between simulations and observations and is thus stricter than the normalized mean error (see Table 1 for equations). Kelley et al. (2013) also suggests to use the Normalized mean error (NME) as a more robust metric than NMSE. NME is based on absolute residuals (NMSE on squared residuals) and thus is especially better suited for variables that can

have very large values and residuals. Additionally, we use the Manhattan metric (MM) proposed by Kelley et al. (2013) for evaluation of vegetation cover. Values for MM less than 1 reflect that the model perform better than the mean value and additionally we show the random model proposed by Kelley et al. (2013, Table 4) for evaluation of vegetation distribution.

Table 2 gives an overview of variables evaluated at the local scale and which measures are used

For the evaluation of time series for crop yields, we employ a simple time series correlation analysis after removing trends with a moving-window detrending method. For comparison with point measurements, we extract the time series from corresponding 0.5° grid cells. These simulated time series may differ in in terms of weather and soil conditions from the actual site as the simulations are based on gridded global data set inputs.

To envisage the degree of agreement between simulated (LPJmL4) and observed (MIRCA2000) sowing dates, we follow Waha et al. (2012) and compute two different metrics: the Willmott coefficient of agreement (W) (Willmott, 1982) and the mean absolute error (ME), both weighted by the



**Table 2.** Overview of variables evaluating LPJmL4, showing measures and references at the local scale.

| | Measure | | | | Reference | |
| | | Standard | | Reference | | |
| Variable | CRMSE | Deviation | Correlation | to figures | Data | Citation |
|---|---|---|---|---|---|---|
| $CO_2$ | | | x | Fig. 1 & 2 | Atmospheric transport | Rödenbeck (2005) |
| NEE | x | x | x | Fig. 3 | FLUXNET | ORNL DAAC (2011) |
| ET | x | x | x | Fig. 7 | FLUXNET | ORNL DAAC (2011) |
| NPP | | | | x | Fig. 4d | | Luyssaert et al. (2007) |
| GPP | | | | x | Fig. 4c | | Luyssaert et al. (2007) |
| BIOMASS | | | | x | Fig. 4a & 4b | | Luyssaert et al. (2007) |
| DISCHARGE | x | x | x | Fig. 8 & | ArcticNET & | |
| | | | | SI | UNH/GRDC | Vörösmarty et al. (1996) |

Centered root mean square error (CRMSE)

crop-specific cultivated area according to (Portmann et al., 2010). For an overview of all metrics used, see Table 1.

## 3   Results and discussion

In the following we compare the standard version LPJmL4, which refers to the experiment LPJmL4-GSI-GlobFIRM. In case of the other experiments we refer to the names defined in Section 2.1.

### 3.1   Vegetation cover

LPJmL4 reproduces the observed vegetation distribution better than the observed mean (MM < 1) and better than the random model (Table 3). Such as the random model, LPJmL4 can best reproduce the distinction between bare soil and vegetated areas (MM = 0.22) and between tree-covered areas and areas without trees (MM = 0.31), but with considerably better scores than the random model (MM = 0.56 and 0.54 respectively). Moreover LPJmL4 simulation results reach the lowest MM scores for the distinction of evergreen vs. deciduous trees (MM = 0.52) and for the distribution and composition of life forms (trees vs. herbaceous vs. bare soil; MM = 0.45), these are substantially better than the random model (MM = 0.87 and 0.88 respectively). The largest improvement of LPJmL4 simulations over the random model are found for the patterns of broadleaved vs. needle-leaved trees (MM = 0.37 for LPJmL4 vs. 0.94 for the random model, see Table 3).

### 3.2   Atmospheric $CO_2$ concentration and NEE

#### 3.2.1   Comparison of simulated NBP to atmospheric $CO_2$ concentration at MLO and BRW

LPJmL4 call well reproduce observed long-term and seasonal dynamics of atmospheric $CO_2$ (Fig. 1 and 2). The long-term trend of atmospheric $CO_2$ is well reproduced in all the different model setups





(Fig. 1), except for the setup for natural vegetation only (LPJmL4-GSI-GlobFIRM-PNV). The experiment with all processes included (LPJmL4-GSI-GlobFIRM) gives the best correlation and trend

reproduction, which suggests that an integral representation of the LPJmL4's features is required to match observations best. Next to land-use dynamics, the inclusion of permafrost dynamics has the strongest effects on the simulated trend (LPJmL4-NOGSI-NOPERRM-GlobFIRM vs. LPJmL4-NOGSI-GlobFIRM). The use of the process-based fire model SPITFIRE leads to small overestimation of the trend in atmospheric $CO_2$ concentrations compared to the other model setups, especially

at MLO. Seasonal variations in atmospheric $CO_2$ can be well reproduced by LPJmL4, especially by the standard setup (LPJmL4-GSI-GlobFIRM) (Fig. 2). The simulation of seasonal variations in atmospheric $CO_2$ content are especially improved by the GSI phenology scheme (LPJmL4-NOGSI-GlobFIRM vs. LPJmL4-GSI-GlobFIRM, Fig. 2 (a)&(b)). All model setups (except LPJmL4-GSI-SPITFIRE) can reproduce the observed strong significant increase in the seasonal $CO_2$ amplitude at

BRW and the weak (but insignificant) increase at MLO (Fig. 2 (c)). These results are in agreement with a previous evaluation of simulated seasonal $CO_2$ changes in LPJmL (Forkel et al., 2016).

Further analysis shows that the standard setup (LPJmL4-GSI-GlobFIRM) can best produce the mean seasonal cycle in MLO, whereas the version that omits land use (LPJmL4-GSI-GlobFIRM-PNV) performs slightly better than this in BRW (Fig. 2). The standard setup (LPJmL4-GSI-GlobFIRM)

can also best reproduce the increase in the seasonal amplitude at BRW, whereas it is the only setup that produces a statistically significant but still very small increase in the seasonal amplitude at MLO, where also observations do not show a statistically significant increase.

### 3.2.2 Comparison of simulated NEE to eddy-flux measurements

We evaluate model performance of simulated NEE from LPJmL4 for temporal and spatial variation

of NEE data from eddy flux measurements, using Taylor diagrams (Taylor, 2001). Stations are sorted from North to South (see Fig. 3) for all NEE measurements available for >3 years and depicted in

**Table 3.** Comparison metric scores for LPJmL4 simulations against observations of fractional vegetation cover data from International Satellite Land-Surface Climatology Project (ISLSCP) II vegetation continuous field (VCF) (Defries and Hansen, 2009).

| Vegetation cover | Manhattan Metric (MM) | |
|---|---|---|
| | LPJmL4 | Random model[*] |
| Life forms | 0.45 | 0.88 |
| Tree vs. non-tree | 0.31 | 0.54 |
| Herb vs. non-herb | 0.42 | 0.66 |
| Bare vs. covered ground | 0.22 | 0.56 |
| Evergreen vs. deciduous | 0.52 | 0.87 |
| Broadleaf vs. needleleaf | 0.37 | 0.94 |

MM suggested by Kelley et al. (2013),[*] values taken from Kelley et al. (2013, Table. 4)



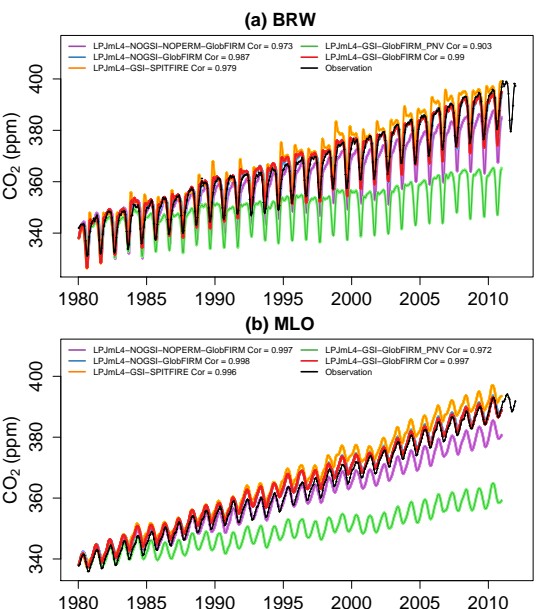

**Figure 1.** Comparison of the atmospheric $CO_2$ concentrations at Mauna Loa (MLO) at the top and Point Barrow (BRW) at the bottom for the different LPJmL4 experiments.

different colors. The model is able to reproduce the mid-latitudes best (represented by yellow over green to light blue colors), with correlation coefficients mostly between 0.4 and 0.9 and standard deviations often within +/-30 % of the reference data. The northernmost regions are well reproduced

at some flux towers, but often with higher standard deviation than in the flux tower data, which means that the simulated time series are largely in phase with but are more variable than the observations. In contrast, the evaluation is comparatively poor for tropical regions, especially the station at Santarém with strong negative correlations (r< -0.6) but realistic standard deviations. For this site, however, Saleska et al. (2003) have already pointed out that the eddy-flux measurements show the

opposite sign compared to tree growth observations and model predictions, which also is the case for LPJmL4. We stress that this evaluation is done for a standard LPJmL4 run and standard input (the LPJmL4-GSI-GlobFIRM as described in Schaphoff et al. (submitted)), i.e. we did not calibrate the model to site-specific conditions and also drive the model with gridded input data rather than the observed soil and weather data at individual stations. More detail for comparisons with eddy-flux tower

measurements for individual locations is supplied in the supplementary material (see SI-Fig. 1-7).





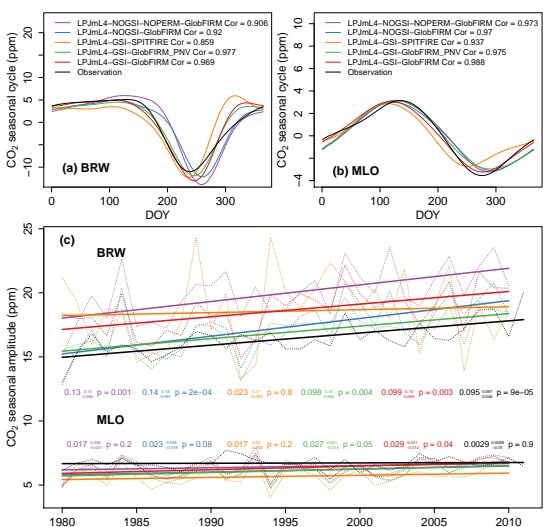

**Figure 2.** Comparison of the atmospheric $CO_2$ concentration at Mauna Loa (MLO) and Point Barrow (BRW) simulated in the different LPJmL4 experiments. Top panel, seasonal cycle; bottom panel, trend of the seasonal amplitude, slope are given for the different LPJmL4 experiments.

## 3.3 Vegetation and soil carbon stocks and vegetation productivity

### 3.3.1 Soil carbon and vegetation carbon stocks

The spatial correlation between simulated and observation-based estimates of soil organic carbon by Carvalhais et al. (2014) is weak (r = 0.29, Table 4) with disagreements in the sub-tropics, where
LPJmL4 simulations substantially underestimate soil carbon stocks, whereas LPJmL4 report much higher soil carbon in the high northern latitudes (>50°N) and lower values for the tropical and temperate zone, compared to Carvalhais et al. (2014) (see SI-Fig. 65). Other estimates by Tarnocai et al. (2009) show much higher carbon content for the permafrost affected areas than the data set of Carvalhais et al. (2014). We thus assume that the disagreement between simulations and the Carvalhais
et al. (2014) data may also result from an underestimation of carbon stocks in the Carvalhais et al. (2014) data.

The comparison of simulated and observation-based assessments of vegetation carbon show a good spatial correlation (r = 0.84, Table 4). The spatial patterns of vegetation carbon stocks are shown in SI-Fig. 87 for simulations and the data product of Carvalhais et al. (2014). While the broad ge-
ographical patterns are in overall agreement with the evaluation data, the absolute values differ in places. Specifically, LPJmL4 simulates much higher biomass (see SI-Fig. 86) for the tropics, and lower biomass between 20 and 40 degrees on the northern and southern hemisphere, where Car-



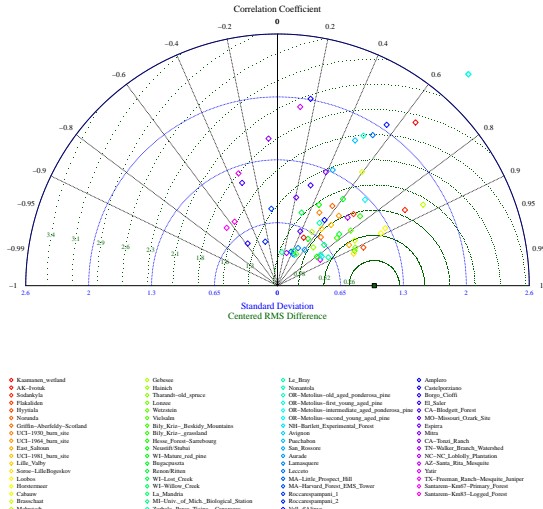

**Figure 3.** Net ecosystem exchange rate measured at eddy flux towers: ORNL DAAC (2011). Available online FLUXNET. Sites (colours) are ordered from north to south.

valhais et al. (2014) show higher values compared to LPJmL4. This is probably due to an overestimation of vegetation carbon in agricultural regions by Carvalhais et al. (2014). The sub-tropical

region where biomass carbon is underestimated corresponds also to the region where LPJmL4 simulations underestimate soil carbon stocks compared to Carvalhais et al. (2014). Also the comparison of aboveground biomass estimates with the data set of Liu et al. (2015) shows a similar spatial pattern of overestimation of vegetation biomass with too high values in boreal and tropical areas. The comparison is complicated by uncertainties in the estimation of belowground biomass (Saatchi et al.,

2011) and the assumed distribution between aboveground and belowground in LPJmL4 simulations, where LPJmL4 assumes that belowground biomass consists of all fine root biomass and one third of all sapwood biomass. The simulation experiments without permafrost dynamics (LPJmL4-NOGSI-NOPERM-GlobFIRM) show a high overestimation of biomass in the high latitudes. Similarly, the inclusion of the GSI phenology substantially reduces the biomass overestimation in comparison to

Carvalhais et al. (2014) and Liu et al. (2015), which is consistent with the finding of Forkel et al. (2014). The consideration of human land use in the simulations improves carbon stock simulations in the temperate zones (SI-Fig. 66). This clearly demonstrates the importance of permafrost, human



**Table 4.** Overview of variables evaluating LPJmL4, showing measures and references at the global scale.

| | | | Measure | | | Reference | |
|---|---|---|---|---|---|---|---|
| | | | spatial | temporal | Visual | | |
| Variable | NME | NMSE | Correlation | Correlation | Comparison | Data | Citation |
| GPP - Av | 0.20 | 0.13 | 0.87 | | Fig. 5 | GPP | Jung et al. (2011) |
| $R_e$ - Av | 0.67 | 0.55 | 0.67 | | Fig. 6 & | | Jägermeyr et al. (2014) |
| | | | | | SI-Fig. 67 | | |
| SoilC - Av | 0.48 | 0.75 | 0.29 | | SI-Fig. 65 | Soil carbon stocks | Carvalhais et al. (2014) |
| VegC - Av | 0.33 | 0.36 | 0.84 | | SI-Fig. 66 | Total Biomass | Carvalhais et al. (2014) |
| FAPAR - I-aMv | 0.17 | 0.13 | 0.63 | Fig. 10a | | MODIS FAPAR | Knyazikhin et al. (1999) |
| FAPAR - I-aMv | 0.18 | 0.15 | 0.59 | Fig. 10b | | GIMMS3g FAPAR | Zhu et al. (2013) |
| FAPAR - I-aMv | 0.21 | 0.20 | 0.69 | Fig. 10c | | VGT2 FAPAR | Baret et al. (2013) |
| ET | 1E-6 | 0.07 | 0.84 | | SI-Fig. 68 | Latent heat flux | Jung et al. (2011) |
| Discharge | | | | | | ArcticNET & | Vörösmarty et al. (1996) |
| Ov | 0.42 | 0.24 | | $R^2 = 0.90$ | | UNH/GRDC | |
| Mav | 0.36 | 0.19 | | $R^2 = 0.92$ | | | |
| I-av | 0.24 | 0.06 | | $R^2 = 0.97$ | | | |

Normalised mean error (NME) and Normalised mean square error (NMSE) as suggested by Kelley et al. (2013); Av – Annual average; I-aMv – Inter-annual-monthly variability; Overall variability – Ov; Monthly average variability – Mav; Inter-annual variability – I-av; Vegetation carbon – VegC; Soil carbon – SoilC.

land use and the GSI phenology for the simulation of the terrestrial carbon cycle, even though the remaining discrepancies warrant further model improvement.

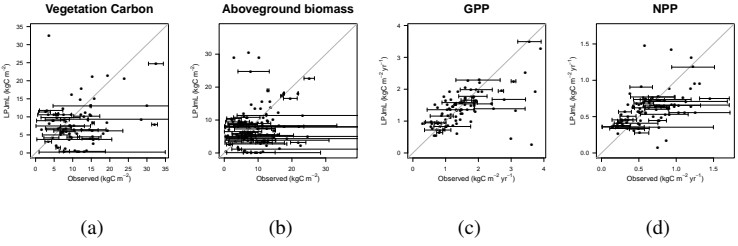

(a)          (b)          (c)          (d)

**Figure 4.** Evaluation of Vegetation carbon (a), aboveground biomass (b), GPP (c), and NPP (d). Observed data are provided by Luyssaert et al. (2007). Bars give the minimum and maximum of the estimation within one 0.5° cell simulated by LPJmL4.

Fig. 4a and 4b compares site data estimation with the representative LPJmL4 grid cell estimation, with an uncertainty range, which comes from the different measurements within one 0.5° grid cell. Both vegetation and aboveground carbon show a slight overestimation of some simulated values, but also some strong underestimation in others. As LPJmL4 calculates a representative mean value of a 0.5° grid cell for all benchmarks, the simulated values should match to the mean values. However, it

can be assumed that measurements are not evenly distributed through the age classes within one grid cell or forest and it remains unclear how representative the measurements for a 0.5° grid cell are.





### 3.3.2 Gross and net primary production (GPP and NPP)

The global estimation of GPP from LPJmL4 (see Fig. 5) is at the upper end of estimates from Jung et al. (2011), whereas the highest divergence can be observed in the tropics, where LPJmL4 estimates

much lower values despite the higher biomass estimations (see Section 3.3). LPJmL4 simulated higher GPP for the temperate and boreal zones than reported by Jung et al. (2011). The different model experiments show similar pattern except for LPJmL4-GSI-GlobFIRM-PNV, which shows lower GPP in the Mediterranean (see Fig. 5).

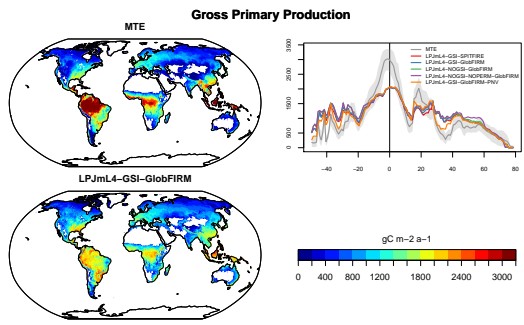

**Figure 5.** The maps (left side) show the spatial pattern of gross primary production (GPP, [$gC\,m^{-2}\,a^{-1}$]) distribution from the standard LPJmL4 simulation against the MTE data (Jung et al., 2011). The graph on the right side shows the latitudinal pattern of evapotranspiration distribution simulated by the different versions of LPJmL4 against data from Jung et al. (2011).

The site data comparison to Luyssaert et al. (2007) shows a good agreement between site mea-

surements and simulated GPP ( see Fig. 4c) and NPP (see Fig. 4d). The overestimation of simulated biomass and the good agreement of NPP and GPP leads to the conclusion that LPJmL4 underestimates mortality. This warrants further investigation why LPJmL4 seems to overestimate global GPP but shows good agreement with site data.

### 3.3.3 Ecosystem respiration ($R_e$)

Comparison of satellite-derived ecosystem respiration with those simulated by LPJmL4 reveals similar spatial patterns (Fig. 6 and SI-Fig. 67). However, LPJmL4 shows higher temperature sensitivities (Fig. 6 (a)) and consistently simulates higher $R_e$ values in high-latitude and subtropical regions (SI-Fig. 67). Since satellite-derived ecosystem respiration is calibrated for FLUXNET data and hence exhibits marginal cross-latitude bias, the discrepancies to LPJmL4 are likely associated either with

LPJmL4 parameterization or with systematic errors in the FLUXNET sampling technique. Additional details and figures are presented in Jägermeyr et al. (2014).





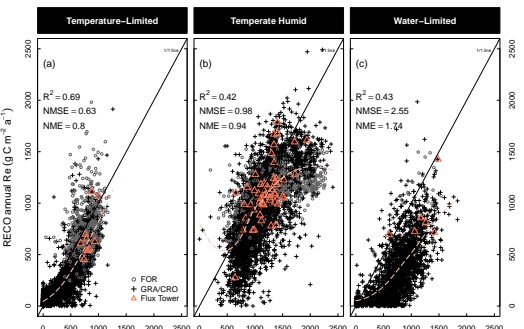

**Figure 6.** Ecosystem respiration ($Re$) evaluation of standard LPJmL4 simulations with satellite-derived estimations from (Jägermeyr et al., 2014). Compared are annual $Re$ sums for all pixels from the displayed extent in SI-Fig. 65, separated by climate type (a)–(c). Dashed lines indicate a polynomial bias curve. Chart symbols are separated for forest (FOR) and grassland/cropland (GRA/CRO) land cover classes.

## 3.4 Water fluxes

### 3.4.1 Evapotranspiration

The spatial distribution of evapotranspiration of LPJmL4 shows a very similar pattern as estimated
by Jung et al. (2011) (Table 4, SI-Fig. 68). It indicates a general underestimation of ET, especially in
the tropics and subtropics, but in most cases within the uncertainty range. This is consistent with the
underestimation of GPP in the tropics (Fig. 5), but not with the general overestimation of vegetation
biomass (SI-Fig. 66). The different experiments show nearly no effects on the simulated evapotranspiration.

At site level, the evapotranspiration fluxes show a good agreement with eddy-flux tower measurements in Fig. 7. LPJmL4 shows good performance in most regions, with correlation coefficients
often larger than 0.6. Especially the northern and temperate stations (red to light blue symbols) show
high correlation with low CRMS. Simulations of tropical and subtropical ET (dark blue to purple
symbols) show weak or even negative correlations coupled with a high CRMS for some stations. We
also provide more detailed time series analyses for the evapotranspiration fluxes of individual sites
in the supplementary material (SI-Fig. 8-16).

### 3.4.2 River discharge stations evaluation

Discharge simulated by earlier LPJmL versions was evaluated before in several studies, also in comparison with other global hydrologial and land surface models (Haddeland et al., 2011). River discharge was evaluated for major catchments globally, also accounting for effects of different precipitation datasets (Biemans et al., 2009) and regionally for the Amazon basin (Langerwisch et al., 2013)
and the Ganges (Siderius et al., 2013).





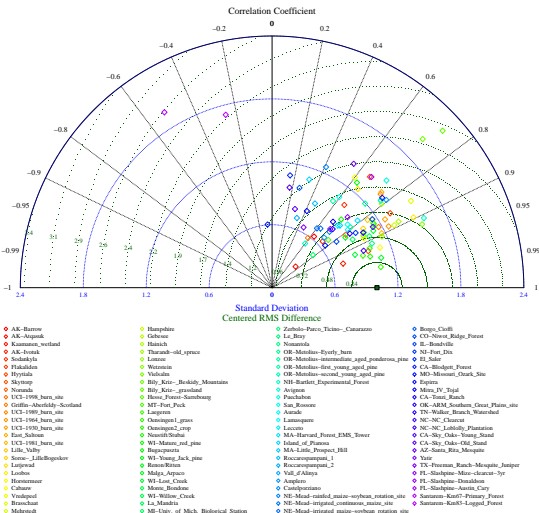

**Figure 7.** Evaporation rate measured at eddy flux towers: ORNL DAAC (2011). Available online FLUXNET. Site locations are ordered from north to south.

Fig. 8 shows the comparison of simulated LPJmL4 and observed river discharge values for all gauges with basin area $\geq 10{,}000\,\mathrm{km}^2$. Here, the most northern (blue) and also most southern (purple)

gauges show good agreement, but overall the picture is mixed with respect to correlation coefficients and standard deviation. For further insights, we provide comparisons for all considered gauges in the supplementary material (SI-Fig. 17-64). For many gauges, the simulated seasonal timing of river discharge (peaks) has improved (see SI-Fig. 17-20) compared to the previous model evaluation of river discharge (Schaphoff et al., 2013), which is mainly a result of the newly implemented GSI-

phenology scheme (Forkel et al., 2014). Especially, the discharge spring peaks in permafrost areas are affected by this improvement. At many gauges, LPJmL4 can reproduce the variability for the whole time series and specially the seasonality, with a high $R^2$ and a NME/NMSE, which implies a better performance than the mean model. The dynamics at gauges in temperate zone (SI-Fig. 47-48, 59) are not well reproduced in the simulations and also the NME/NMSE show high values in

contrast to gauges in the subtropics and tropics (SI-Fig. 62-64), which typically show high $R^2$ and low NME/NMSE.

The evaluation at the global aggregation (computed for all stations and than averaged) shows very high agreement between observed and modelled discharge (see Table 4). Both the explained





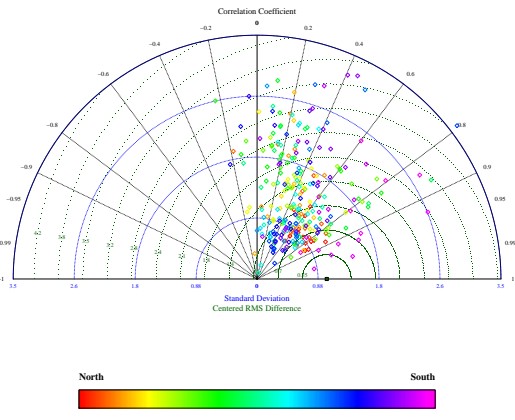

**Figure 8.** Comparison of simulated discharge with 287 gauges provided by ArcticNET and UNH/GRDC. Stations with basin area $\geq 10,000\,\mathrm{km}^2$ are taken into account. Gauges are ordered from north to south (see legend color).

variance ($R^2$) and the NME/NMSE contribute to the good performance of the simulated discharge.

The constant flow velocity in all rivers, as assumed in LPJmL4 simulations, could be varied by river for further model improvement, especially for the timing in flat areas where wetland dynamics may play an important role.

### 3.4.3 Irrigation withdrawal and consumption

Global estimates of irrigation water withdrawal ($W_d$: 2577 km$^3$) and consumption ($W_c$: 1299 km$^3$)
agree well with previous studies. Reported $W_d$ values for the period 1998-2012 are 2722 km$^3$ (FAO, 2014), and modelling results range from 2217 to 3185 km$^3$ (Döll et al., 2014; Wada and Bierkens, 2014; Döll et al., 2012; Alexandratos and Bruinsma, 2012; Wada et al., 2011; Siebert and Döll, 2010). $W_c$ estimatations range between 927 and 1530 km$^3$ (Chaturvedi et al., 2015; Döll et al., 2014; Hoff et al., 2010). Döll et al. (2012) finds that 1179 km$^3$ (1098 km$^3$ in Wada and Bierkens (2014))
relate to surface water and additional 257 km$^3$yr$^{-1}$ from groundwater resources. LPJmL4 does not account for fossil groundwater extraction nor desalination. However, previous studies show that 80% of groundwater withdrawals are recharged by return flows (Döll et al., 2012). It is thus plausible that studies accounting for (fossil) groundwater reach $W_d$ estimates somewhat higher than in LPJmL4. Naturally, irrigation water estimates are associated with uncertainties in the precipitation input em-
ployed (Biemans et al., 2009). A representation of multiple cropping systems in LPJmL4 (Waha et al., 2013) and corresponding growing seasons (Waha et al., 2012) could also help to improve





water withdrawal and consumption estimates and eventually river discharge, especially in tropical areas.

Simulated irrigation efficiencies are difficult to compare with observations due to inhomogeneous definitions and field measurement problems. Yet, in SI-Table 1 we relate our results to comparable literature. Our simulations meet indicative estimates of Brouwer et al. (1989) at global level. Sauer et al. (2010) provide another independent estimate of field efficiency with global average values of 42%, 78%, and 89% for the three irrigation types, respectively. Our estimates agree well with these numbers globally and regionally, even though there are some regional patterns that are not represented in our results. Sauer et al. (2010), for instance, find lower surface irrigation efficiencies in Middle East, North Africa (MENA) and sub-Saharan Africa (SSA). We simulate above-average efficiencies in MENA and particularly low ones in South Asia, which is both supported by Rosegrant et al. (2002) and Döll and Siebert (2002). Overall, the evaluation of the irrigation model in LPJmL4 demonstrates that it is well in line with reported patterns and yet it comes with much more detail depths with respect to process representation and spatio-temporal resolution than these.

### 3.5 Permafrost distribution and active-layer thickness

The current permafrost distribution and the active-layer thickness (Fig. 9) is well represented by the LPJmL4 model compared to independent studies (Brown et al., 1998, 2000) . LPJmL4 is able to reproduce the distribution of permafrost and the measured active-layer thickness in most grid cells. The spatial distribution of greater thaw depth from north to south is simulated well by the model.

### 3.6 Fire

#### 3.6.1 Burnt area

Simulated fractional area burnt is largest in the seasonal dry tropics and temperate regions in all model versions and smallest in cold or wet environments (SI-Fig. 69). However, maximum fractional burnt area does not exceed 0.0625 in tropical and subtropical savannah and shrubland areas when the Glob-FIRM model is applied. It is comparable to GFED4 and CCI estimates only in South America, while in other tropical regions GFED4 (Giglio et al., 2013) and CCI reports fractional burnt area between 0.125 and 0.75 (SI-Fig. 69). In these regions, fractional burnt area simulated by the SPITFIRE model is overestimated with values between 0.25 and 1, specifically in the southern hemispheric Africa and northern Australia. SPITFIRE is very sensitive to vegetation, thus fuel composition where homogeneous C4 grasslands can lead to an overestimation of simulated area burnt which is specifically the case for seasonally dry South America and the Indian subcontinent. LPJmL4-GSI-SPITFIRE captures the distribution of fractional burnt area much better than LPJmL4-GSI-GlobFIRM which is too homogeneous in its response.



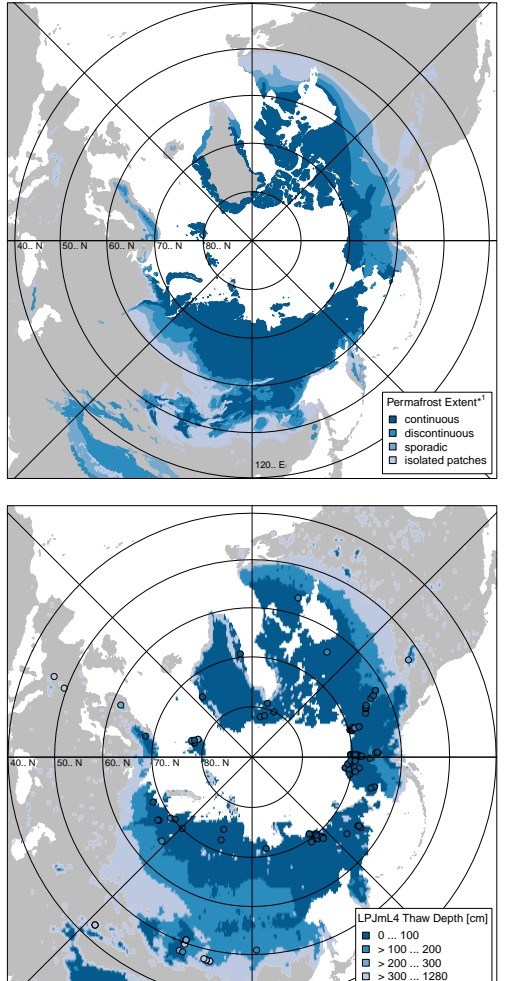

**Figure 9.** Observed and simulated permafrost distribution and active layer thickness. Top, contemporary permafrost extent according to the IPA Circum–Arctic Map of Permafrost (*[1] Brown et al. (1998)). Bottom, LPJmL4-simulated active-layer thickness compared to the *[2] CALM station data means both for the observation time 1991-2009 (http://www.gwu.edu/ calm/; Brown et al. (2000)). The colour scheme used at the bottom are the same for simulated thaw depth and Circumpolar Active Layer Monitoring (CALM) data.

In contrast, LPJmL4-GSI-SPITFIRE better captures the very small fractions reported for the wet
tropical forests which is better comparable to GFED4. Here, the approach to simulate fire risk based
on the climatic fire danger index instead of deriving a fire probability from the top-soil soil moisture





is of great advantage in these regions. While LPJmL4-GSI-GlobFIRM simulates a relatively homo-geneous spatial distribution of fractional burnt area in temperate and boreal forest regions, LPJmL4-GSI-SPITFIRE underestimates fractional burnt area in these biomes. LPJmL4-GSI-GlobFIRM un-derestimates fractional burnt area in the temperate steppe regions, whereas LPJmL4-GSI-SPITFIRE manages to spatially capture the burning conditions in these biomes, even though the total amount is overestimated. The phenology module in LPJmL4 has no effect on fractional burnt area simulated by LPJmL4-GSI-GlobFIRM, whereas including permafrost increases burnt area in the circumboreal region, specifically in Siberia, even though the spatial effect is too homogeneous.

### 3.6.2 Fire effects on biomass and vegetation distribution

Both fire model approaches simulate a comparable latitudinal distribution of biomass starting from the wet tropics towards dry and colder areas in the North and South. Both model version simu-late comparable values in the wet tropics around the equator and capture the gradient to seasonal dry tropics in the North (until 10°N) and South (until 20°S). The overestimation of burnt area in tropical savannahs around 20°N in LPJmL4-GSI-SPITFIRE leads to an underestimation in simu-lated biomass compared to the other LPJmL4 experiments. The consideration of permafrost and fire dynamics is required to reproduce observed vegetation biomass values in boreal regions.

### 3.6.3 Global biomass burning

The modelling errors in fractional area burnt compensate in different ways in each fire model. SPIT-FIRE simulates global biomass burning values of 2.7 PgC p.a. on average between 1996-2005 which is comparable to the 2.33 PgC p.a. (Randerson et al., 2015). Here, overestimations of burnt area in tropical savannahs and underestimations in boreal forests compensate each other. Glob-FIRM sim-ulates more fires in boreal regions, but less spatially pronounced as in GFED4, but underestimates fractional burnt area in the subtropics and tropics. Glob-FIRM therefore estimates global biomass burning by 2.8 PgC $yr^{-1}$, similar to SPITFIRE.

### 3.7 Fraction of absorbed Photosynthetically Active Radiation- (FAPAR) and Albedo

Evaluations against multiple satellite datasets of FAPAR have already shown that LPJmL-GSI can well reproduce the seasonality of FAPAR and the inter-annual variability and trends in the start and end of growing season within observational uncertainties (Forkel et al., 2015). LPJmL4 shows a high spatial correlation with correlation coefficients between 0.6 and 0.71 for PEAK-FAPAR. It shows also a good agreement with the temporal variations Fig. 10a-10c. Large parts of the wet trop-ics display a negative correlation between simulated and observed FAPAR, which may explain the phase-offset in the dynamics of NEE at the station Santarém. However, in these regions also the dif-ference between datasets are large which is caused by the limitations of optical satellite observations in regions with permanent cloud cover (Forkel et al., 2015).

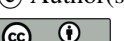



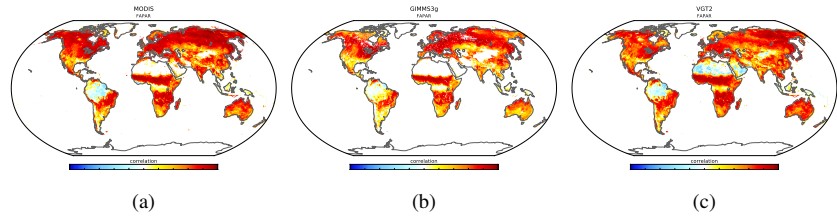

|  (a) | (b) | (c) |

**Figure 10.** Evaluation of FAPAR for different data sources MODIS (a), GIMMS (b), and VGT2 (c).

LPJmL4 reproduces the global patterns of annual peak FAPAR (Fig. 11) well. Especially, in northern latitudes and in the tropics, LPJmL4 is within the range of the FAPAR datasets. However, LPJmL4 overestimates peak FAPAR especially in middle and low latitudes which originates form an
overestimation of FAPAR in semi-arid regions. LPJmL4 reproduces well the temporal dynamic of FAPAR in most climate regions with very high correlations between simulated and observed FAPAR in temperate and boreal climates (climate regions Cf and D*) and with medium to high correlations in semi-arid climate regions (e.g. Am, As, Aw, Bsh, Bsk, Cs in SI-Fig. 70). LPJmL4 and the observational datasets show low correlations in wet tropics (Af) and in winter-dry temperate climates
(Cw).

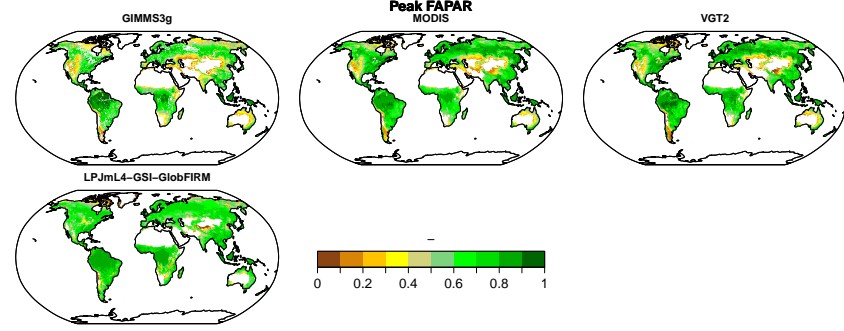

**Figure 11.** FAPAR mean annual peak comparison with 3 different remote sensing products.

LPJmL4 overestimates albedo in all regions (SI-Fig. 71. The temporal dynamic of snow-free albedo was well reproduced in cold steppes (climate region BSk) and in boreal regions (climate regions D*). The correlation between simulated and observed albedo is poor in tropical semi-arid and temperate climates (e.g. As, Aw, Cs, Cf). This is likely caused by soil moisture-induced changes
in soil and background albedo, which has a great effect on soil reflectance (Lobell and Asner, 2002) outside the vegetation season. Such changes are not considered in LPJmL4.



### 3.8 Agriculture

#### 3.8.1 Crop yields variability

The evaluation of simulated crop growth and yield can be assessed at individual sites if the model
is used as a point model as in different model intercomparison simulations (Asseng et al., 2013;
Bassu et al., 2014; Kollas et al., 2015; Asseng et al., 2015) where reference data are available for
end-of-season properties (most importantly: crop yield) as well as within-season dynamics (e.g.
development of leaf area index (LAI)). The crop yield simulations of LPJmL were evaluated in
the framework of the Agricultural Model Intercomparison and Improvement Project (AgMIP) for
wheat, maize, rice and soybean by (Müller et al., 2017). They find that the performance of LPJmL is
similar to that of the other gridded crop models in that model ensemble (n = 14). We here supplement
the model evaluation with time series correlation analyses for the ten top-producing countries for
all crops implemented in LPJmL4 (Schaphoff et al., submitted). Results are portrayed in Fig. 12,
except for field peas where no spatial data on crop-specific harvested areas exists for aggregation
to national yield time series (Porwollik et al., 2016). As national yield levels are roughly calibrated
in standard LPJmL simulations (Fader et al., 2010), a comparison of the mean bias is not providing
insights on model performance. For a more comprehensive evaluation of LPJmL's performance in
yield simulations, see Müller et al. (2017).

The agreement between simulated and observed yields is not only dependent on model perfor-
mance, but also on the aggregation mask used (Porwollik et al., 2016), assumptions on management
and model parametrization (Folberth et al., 2016a), soil parameters (Folberth et al., 2016b) and
weather data inputs (Ruane et al., 2016). LPJmL4 yield simulations are typically correlated with
national yield statistics (FAO, 2014) for some of the 10 top-producing countries for each crop, but
only for one of these for cassava (Brazil) and sugarcane (China) (Fig. 12 and supplementary material
Fig. 66-74 for the other crops).

#### 3.8.2 Biomass yield

For the purpose of this evaluation, irrigated and rainfed biomass plants were simulated to grow glob-
ally, wherever biophysical conditions allow sustained growth. The averaged simulated yields for the
16-year period (1994–2009) were compared to reported biomass yields of switchgrass, miscanthus,
poplar, willow and eucalyptus plantations on experimental test-sites located in the respective grid
cell (Fig. 13). It shows that simulated yields are mostly within the range of observations for miscant-
hus, poplar, willow and eucalyptus, but mostly overestimates switchgrass productivity. Management
options for BFTs implemented in LPJmL4 are limited to irrigation management (rainfed and fully
irrigated), because plant species and plantation characteristics (e.g. sapling size and crop spacing)
are parametrised as a constant scenario setting and were not varied here. The differences between
rainfed and irrigated biomass yield simulations are depicted as vertical error bars in Fig. 13. The



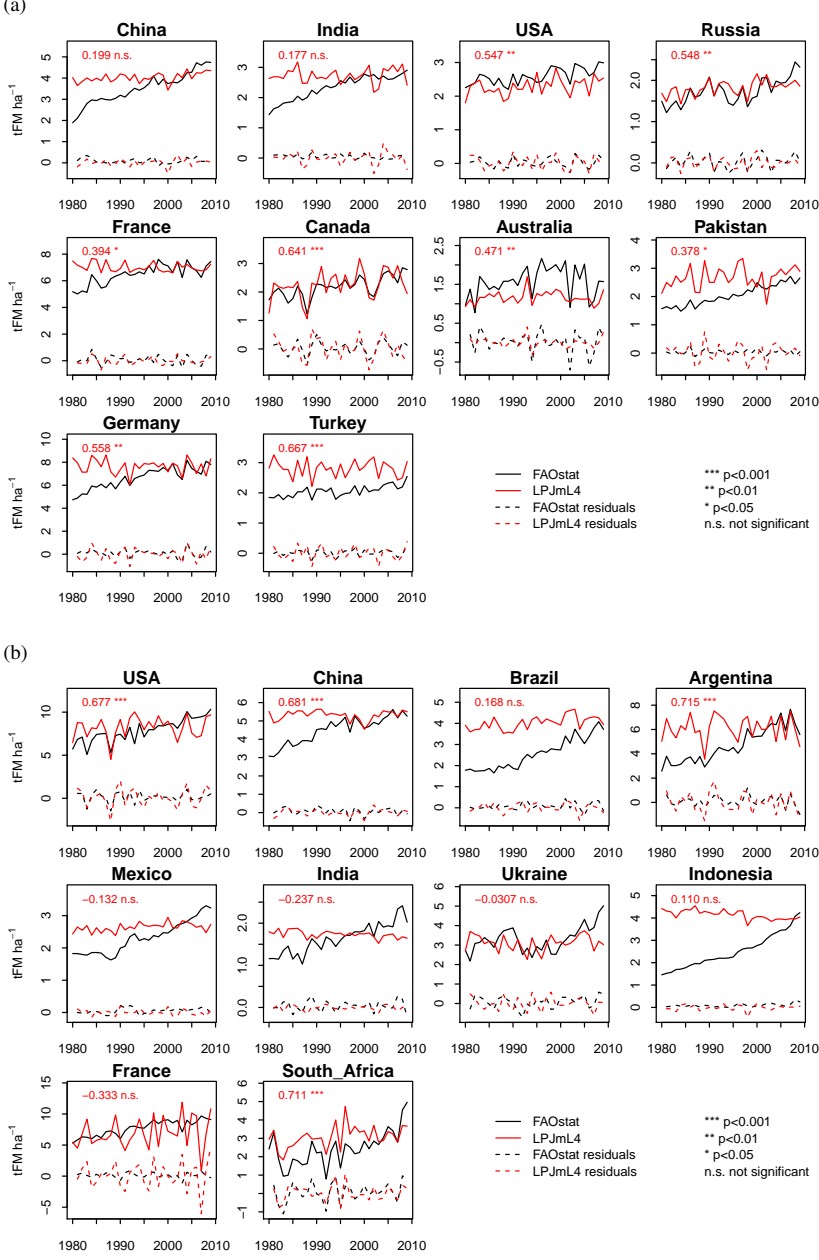

**Figure 12.** Evaluation of simulated yield variability for wheat (a) and maize (b) in comparison to FAO-data
(FAOSTAT).



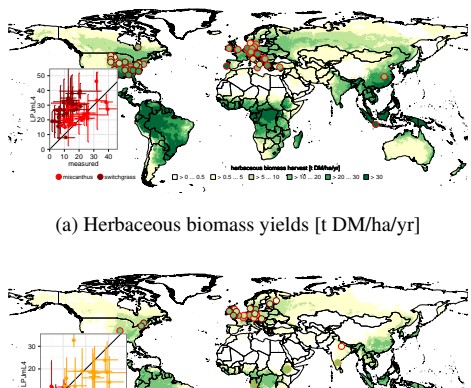

(a) Herbaceous biomass yields [t DM/ha/yr]

(b) Woody biomass yields [t DM/ha/yr]

**Figure 13.** Map of simulated biomass yields by LPJmL4 from rainfed herbaceous (a) and woody (b) BFTs (averages 1994–2009). Dots indicate the location of the experimental sites and measured yield, with colours scaled to map colours. Scatterplots compare observed and simulated yields in the respective grid cells. Model uncertainty is derived from simulations with and without irrigation. Observation uncertainty reflects dependencies on plantation management (adapted from Heck et al. (2016)).

range of rainfed vs. fully irrigated biomass yields represent an approximation of management uncertainty, because simulated yields depend strongly on water availability. Nevertheless the simulated yield range is likely to represent an optimal field management for rainfed resp. irrigated plantations

as nutrient limitations are not taken into account in these simulations.

### 3.8.3   Month of sowing

The average mean error (ME) for all crops globally is smaller than two months, with the exception of pulses (Table 5). For wheat (excl. Russia), millet, rice, sunflower and sugar beet, the agreement between simulated and observed timing of sowing is higher, with a difference of about one month.

The Willmott coefficients (W) are high indicating good agreement between observations and simulations (W > 0.85) for all crops except pulses, sugar beet and groundnut. Both measures indicate closer agreement for pulses, groundnut, sunflower and rapeseed in temperate regions (Waha et al., 2012). Poor agreement, with differences between simulated and observed sowing dates of more than five months, is found for maize and cassava in Southeast Asia and China (for maize in East Africa),

for wheat in Russia, for pulses in Southeast Asia, India, West and East Africa, the south-east region of Brazil and southern Australia, for groundnut in India and Indonesia, and for rapeseed in southern Australia and southern Europe (for wheat Fig. 14; for the other crops SI-Fig. 65-74). Divergences



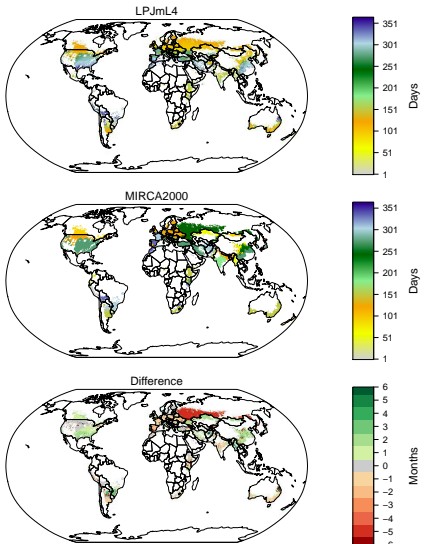

**Figure 14.** Evaluation of sowing dates of wheat: (from top to bottom panel) simulated (LPJmL4) sowing date, observed (MIRCA2000) sowing date and difference between simulated and observed sowing date. Green colours (red colours) in the difference map indicate that simulated sowing dates are too late (too early) compared to observations. White colours indicate crop area with less than 0.001% of the grid cell area. Regions without seasonality are not shown.

are also substantial for crops growing in the southern part of the Democratic Republic of Congo, in Indo-China and in tropical climates.

There are several reasons for these disagreements between sowing dates simulated solely using climate data and the global crop calendar, please see Waha et al. (2012) for a more detailed discussion. Firstly the crop varieties in the crop calendar and simulated here differ, i.e. spring and winter varieties of wheat and rapeseed in temperate regions (e.g. in Russia). Secondly, multiple cropping in tropical regions with high cropping intensity and complex cropping systems is not considered here.

Thirdly, we use of only one global temperature threshold for simulating sowing temperatures, which is known to vary between regions and lastly, there are other uncertainties in our method of simulating sowing dates and in the global crop calendar we use for comparison. We are also neglecting important factors such as the availability of labour and machinery, social customs, markets and prizes, the demand for certain agricultural products at certain times in the year.

The comparison to the global crop calendar, however, shows that close agreement between simulated and observed sowing dates can be achieved with purely climate-driven rules for large parts of





**Table 5.** Indices of agreement between simulated (LPJmL4) and observed (MIRCA2000) sowing dates.

| Crop | All cells | | | Precipitation seasonality | | | Temperature seasonality | | |
| --- | --- | --- | --- | --- | --- | --- | --- | --- | --- |
| | W [-] | ME [days] | N | W [-] | ME [days] | N [%] | W [-] | ME [days] | N [%] |
| wheat | 0.87 | 44 | 13962 | 0.86 | 40 | 15 | 0.87 | 44 | 85 |
| rice | 0.90 | 25 | 4995 | 0.90 | 24 | 82 | 0.87 | 28 | 18 |
| maize | 0.88 | 37 | 16333 | 0.89 | 37 | 48 | 0.85 | 36 | 52 |
| millet | 0.89 | 17 | 7851 | 0.92 | 16 | 63 | 0.89 | 31 | 37 |
| pulses | 0.63 | 69 | 14712 | 0.61 | 80 | 48 | 0.84 | 37 | 52 |
| sugarbeet | 0.37 | 19 | 2918 | 0.24 | | | 0.37 | 19 | 100 |
| cassava | 0.93 | 51 | 6082 | 0.93 | 51 | 83 | 0.95 | 57 | 17 |
| sunflower | 0.92 | 25 | 5876 | 0.87 | 45 | 22 | 0.93 | 22 | 78 |
| soybean | 0.94 | 36 | 8259 | 0.94 | 35 | 31 | 0.92 | 36 | 69 |
| groundnut | 0.77 | 34 | 5642 | 0.71 | 36 | 81 | 0.96 | 20 | 19 |
| rapeseed | 0.86 | 49 | 5680 | 0.36 | 135 | 13 | 0.92 | 37 | 87 |
| wheat excl. Russia | 0.94 | 30 | 11511 | 0.86 | 40 | 18 | 0.94 | 29 | 82 |

Mean absolute error (ME) and the Willmott coefficient of agreement (W)

the earth for wheat, rice, maize, millet, soybean and sunflower, as well as for pulses and groundnut
in temperate regions. For about 75% of the global cropping area the difference between simulated
and observed sowing dates is two months and with the exception of cassava and rapeseed 80% of
the crop area displays a difference of only one month which is the minimum difference possible as
the crop calendar reports monthly sowing dates.

## 4  Conclusions

This article provides a comprehensive evaluation of the now launched version 4.0 of the LPJmL
DGVM that includes an operational representation of agriculture. Unique in its combination of fea-
tures, the LPJmL4 model enables simulation of carbon and water fluxes linked to the dynamics of
both natural and agricultural vegetation in a single, internally consistent frameworks. By following
suggestions for objective intercomparative benchmarking systems of multiple models with dedicated
software (Abramowitz, 2012; Kelley et al., 2013; Luo et al., 2012), the evaluation takes into account
a number of performance metrics, diagnostic plots and a broad range of fundamental model fea-
tures. This work thus goes well beyond earlier evaluations of DGVMs (see Kelley et al. (2013)) and
of model evaluations published for earlier versions of LPJmL or its modules.

Pending major model improvements — anticipated as part of forthcoming LPJmL versions — are
the incorporation of a scheme for calculating groundwater recharge and storage, the representation of
nitrogen cycling for both natural and agricultural landscapes, consideration of ozone effects on plants
(Schauberger et al., submitted) and of soil degradation, representation of wetlands with associated
methane emissions, the continuous refinement of crop parameterization including multi-cropping



and other management forms, and possibly a revised implementation of soil moisture (following e.g.
Evaristo et al. (2015)) and stomatal conductance (following e.g. Lin et al. (2015)). As such improve-
ments are expected to have significant effects e.g. on plant production, carbon and water fluxes –
thus influencing overall model performance – any future LPJmL version will routinely be subjected
to the evaluation protocol used here and, if applicable, tested against other standardized inter-model
benchmarks (including participation in model intercomparisons with evaluation of single compo-
nents such as in Hattermann et al. (2017)). Such continued model maintenance and benchmarking
shall also keep pace with recent developments in observational and experimental data, ideally sup-
porting identification of key uncertainties in model performance (see Medlyn et al. (2015); Smith
et al. (2016)).

Besides identifying features for future model improvement, we here demonstrate adequate perfor-
mance of the LPJmL4 DGVM in terms of the simulation of long-term averages and also the temporal
dynamics across biogeochemical, hydrological and agricultural processes. This unique capacity ren-
ders the LPJmL4 model suitable for process-based analyses of biosphere dynamics including assess-
ments of multi-sectoral impacts of climate change or other anthropogenic earth system interference.

## 5   Code and data availability

As in the companion paper Part I, we will make the model code of LPJmL4 through a Gitlab repos-
itory publicly available. Additionally, we will provide data from the model simulations used here in
a research data repository (see http://dataservices.gfz-potsdam.de/portal/), including all experiments
conducted for Part II. Evaluation data availability is described in section 2.2.

*Acknowledgements.* This study was supported by the German Federal Ministry of Education and Research's
(BMBF's) "PalMod 2.3 Methankreislauf, Teilprojekt 2 Modellierung der Methanemissionen von Feucht- und
Permafrostgebieten mit Hilfe von LPJmL", (FKZ 01LP1507C). M.F. was funded by the TU Wien Wissenschaft-
spreis 2015 awarded to Wouter Dorigo. This work used eddy covariance data acquired and shared by the
FLUXNET community, including these networks: AmeriFlux, AfriFlux, AsiaFlux, CarboAfrica, CarboEu-
ropeIP, CarboItaly, CarboMont, ChinaFlux, Fluxnet-Canada, GreenGrass, ICOS, KoFlux, LBA, NECC, OzFlux-
TERN, TCOS-Siberia, and USCCC. The ERA-Interim reanalysis data are provided by ECMWF and processed
by LSCE. The FLUXNET eddy covariance data processing and harmonization was carried out by the European
Fluxes Database Cluster, AmeriFlux Management Project, and Fluxdata project of FLUXNET, with the support
of CDIAC and ICOS Ecosystem Thematic Center, and the OzFlux, ChinaFlux and AsiaFlux offices.We thank
the coordinators of the Circumpolar Active Layer Monitoring ( CALM) program for providing thaw depth, the
National Snow & Ice Data Center for providing Circum-Arctic Map of Permafrost, and the R-ArcticNET for
discharge data. Furthermore. we thank Jena-BGI data for providing GPP, latent heat flux, total biomass and
soil carbon data. We also thank the providers of the evaluation data of FAPAR (GIMMS3g FAPAR, VGT2
FAPAR, MODIS FAPAR) and the remote sensing data of GFED4 and CCI for evaluating fractional burnt area.



MODIS C5 albedo time series data product was retrieved from the online Data Pool, courtesy of the NASA
Land Processes Distributed Active Archive Center (LP DAAC), USGS/Earth Resources Observation and Science (EROS) Center, Sioux Falls, South Dakota, . We thank the Climatic Research Unit for providing global
gridded temperature input, the Global Precipitation Climatology Centre for providing precipitation input and
the coordinators of ERA-Interim for providing shortwave downward radiation and net downward longwave
radiation. Furthermore, we thank the coordinators of MIRCA2000 for providing land use input.





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
