# Peer review of "LPJmL4 – a dynamic global vegetation model with managed land: Part II – Model evaluation"

_Geoscientific Model Development, 2017_

## Short Comment (SC1) · 5 Aug 2017

see https://editor.copernicus.org/index.php/gmd-2017-145-SC1.pdf?_mdl=msover_md&_jrl=365&_lcm=oc108lcm109w&_acm=get_comm_file&_ms=59866&c=127647&salt=1510553859679472192

---

## Author Comment (AC1) · 17 Aug 2017

see response here:

https://editor.copernicus.org/index.php/gmd-2017-145-AC1-print.pdf?_mdl=msover_
md&_jrl=365&_lcm=oc108lcm109w&_acm=get_comm_print_file&_ms=59866&c=
128157&salt=16837466451869795907
* * *

---

## Referee Comment (RC1) · Anonymous Referee #1 · 14 Sep 2017

General comments: This paper presents a comprehensive evaluation of the new model version, LPJmL4, which shows the strengths and shortcomings of the model and identifies the need of further model improvement. This evaluation mainly focuses on stocks and flows of carbon and water in natural and managed ecosystems at various temporal and spatial scales, providing an elegant example of DGVM assessment.

Specific comments: I have two concerns on the manuscript.

1. The increasing crop production trend as mainly driven by the agricultural Green Revolution did not seem to simulate well, for wheat, e.g., China, India, France, Pakistan and Germany in Fig.12a, and for maize, most countries in Fig.12b, and for rice, most countries in SI-Fig.72. Are there any representations of the Green Revolution in the model like other models (Gray et al., 2014; Zeng et al., 2014)? And/or do the driv-

ing datasets include management practices (high-yield variety selection, irrigation and fertilizer and pesticide application)? Please provide more details in Section 2.1 and/or discuss this in Section 3.8.

2. The scale mismatch problem between site observed data and model simulated results as mentioned in Line 122-132 makes the comparison of vegetation carbon and aboveground biomass in Fig.4a-b much more difficult. One possible method to avoid such mismatch may be to calibrate and validate the model using site specific climate, edaphic, vegetation and management datasets. With site and/or regional calibrated parameters, the comparison between observed and simulated results would make more sense.

Technical corrections and some minor comments:

1. Please introduce each abbreviation in the manuscript text after it is first used. For example, FAPAR was firstly used in Line 29, not Line 106; GPP firstly in Line 117, not in 163;NEE in Line 108 is better after net ecosystem exchange and could avoid in Line 181. Please also check other abbreviations.

2. FAOstat in Line 110 may be better for FAOSTAT;

3. In Line 189 "a empirical model" should be an empirical model;

4. The citation for HWSD data (Line 189) is better for: FAO/IIASA/ISRIC/ISSCAS/JRC, 2012. Harmonized World Soil Database (version 1.2). FAO, Rome, Italy and IIASA, Laxenburg, Austria;

5. There is double "a" in Line 201, double "in" in Line 313, delete one;

6. CALM and IPA are firstly used in Line 219; GFED and CCI in 226;

7. A period should be at the end of Line 309;

8. "soil organic carbon" in Line 188 can be short for SOC;

[Figure]

9. Soil carbon pool can also compare to (Tian et al., 2015);

10. Line 391 SI-Fig.86 might be SI-Fig.66a for biomass? Also check other SI-Figs.

11. Add linear regression coefficients of slope, intercept, R square (R2), P value and root mean square error (RMSE) for Fig. 4; For Fig.4c, to provide a side by side sub-plot of GPP from MTE against observed data (in SI) can be beneficial.

12. Line 501 MENA is better separated for ME and NA.

13. Section 3.5 may be too short. Add more details on permafrost area and active-layer depth dynamics.

14. Line 546-547 used PgC p.a., and Line 551 PgC yr-1, please keep consistency.

15. Please provide full name for "FM" in Fig.12 caption and "resp." in Line 614.

References:

Gray, J. M., Frolking, S., Kort, E. A., Ray, D. K., Kucharik, C. J., Ramankutty, N., and Friedl, M. A.: Direct human influence on atmospheric CO2 seasonality from increased cropland productivity, Nature, 515, 398-401, 2014.

Tian, H., Lu, C., Yang, J., Kamaljit, B., Huntzinger, D. N., Schwalm, C. R., Michalak, A. M., Robert, C., Philippe, C., and Daniel, H.: Global patterns and controls of soil organic carbon dynamics as simulated by multiple terrestrial biosphere models: Current status and future directions, Global Biogeochemical Cycles, 29, 775, 2015.

Zeng, N., Zhao, F., Collatz, G. J., Kalnay, E., Salawitch, R. J., West, T. O., and Guanter, L.: Agricultural Green Revolution as a driver of increasing atmospheric CO2 seasonal amplitude, Nature, 515, 394-397, 2014.

---

## Referee Comment (RC2) · Anonymous Referee #2 · 21 Sep 2017

Comments on "LPJmL4 – a dynamic global vegetation model with managed land: Part II – Model evaluation"

General comments

In this manuscript, the authors presented results of model benchmarking of their newly developed model, LPJmL4. They used many contemporary observational (then independent) data for the benchmarking, spanning a wide range of model aspects such as productivity, hydrology, and agriculture. Through this attempt, they clarified characteristics of LPJmL4 in comparison with other models and previous versions. This benchmarking focused on site to global features and so did not go into details of ecological vegetation dynamics, plant physiology, soil biogeochemistry, and human management. Nevertheless, such benchmarking is an increasingly important task for model intercomparison, and this study is a good attempt.

The manuscript is, frankly speaking, quite long, although this is the second part of the full length of their work. Result description of each examined variable may be shortened to some extent (not mandatory). Overall, as a benchmarking paper, this manuscript is reasonably organized, and I found no logical fault.

Specific comments

1. Line 40: I agree that benchmarking became more and more important and several standardized systems have been proposed. As an example, I suggest referring the iLAMB (https://www.ilamb.org/) as a representative system.

2. Line 61: Harris (2015) does not appear in References.

3. Line 65: Please give full words for NCEP.

4. Line 64: As long as I know, all meteorological forcing variables are available from ERA-interim (or other appropriate dataset). By using the single dataset, you could conduct more comprehensive simulations with higher integrity. Why did you use different datasets?

5. Line 141: This sentence could be removed or merged to other sentences.

6. Line 208: Please add a reference to the FLUXNET data base.

7. Line 231: Just confirmation. You did not use any data of solar-induced chlorophyll fluorescence (SIF) for benchmarking FAPAR and GPP. OK? Because SIF is increasingly used in such benchmarking, I suggest at least referring the use of SIF in your forthcoming study.

8. Line 310: "For" to "for"

9. Line 324: What do you mean for "observed mean" of vegetation distribution?

10. Line 336: Remove "call". OK?

[Figure]

11. Line 389: SI-Fig.87 should be SI-Fig.66.

12. Line 393: Can you explain why such overestimation occurred in vegetation biomass of Carvalhuis et al. (2014)?

13. Line 418: Why did not you provide global values of GPP and NPP? You did so for irrigation and biomass burning emission.

14. Line 435: Figure 6. Please add a title and units for x-axis.

15. Line 546: "Pg C p.a." to "Pg C yr-1"

16. Line 562: Units and numbers of each color scale are difficult to read.

17. Line 564: "form" to "from".

18. Line 619: Maybe, "beans" is more popular than "pulses" (if correct).

---

## Author Response (AR1)

Reply to referee #1

> We thank Anonymous Referee 1 for the thoughtful comments and suggestions and for their careful reading of the manuscript. Line numbers refer to the marked-up version of the manuscript.

*"General comments: This paper presents a comprehensive evaluation of the new model version, LPJmL4, which shows the strengths and shortcomings of the model and identifies the need of further model improvement. This evaluation mainly focuses on stocks and flows of carbon and water in natural and managed ecosystems at various temporal and spatial scales, providing an elegant example of DGVM assessment. "*

*"Specific comments: I have two concerns on the manuscript.*
*1. The increasing crop production trend as mainly driven by the agricultural Green Revolution did not seem to simulate well, for wheat, e.g., China, India, France, Pakistan and Germany in Fig.12a, and for maize, most countries in Fig.12b, and for rice, most countries in SI-Fig.72. Are there any representations of the Green Revolution in the model like other models (Gray et al., 2014; Zeng et al., 2014)? And/or do the driving datasets include management practices (high-yield variety selection, irrigation and fertilizer and pesticide application)? Please provide more details in Section 2.1 and/or discuss this in Section 3.8."*

> Indeed, the intensity and management of crop production are static, so that any intensification driven by inputs (fertilizers, pest control) and/or new varieties (green revolution) are not reproduced. The comparison against FAO data works thus with de-trended data, as typically done for global-scale crop models (see e.g. Müller et al. 2017). We have added more details on the model setup of crop production in the LPJmL4 model (see L.: 79-82)  and a respective extension in the section 3.8.1 (L.: 633-635).

*"2. The scale mismatch problem between site observed data and model simulated results as mentioned in Line 122-132 makes the comparison of vegetation carbon and aboveground biomass in Fig.4a-b much more difficult. One possible method to avoid such mismatch may be to calibrate and validate the model using site specific climate, edaphic, vegetation and management datasets. With site and/or regional calibrated parameters, the comparison between observed and simulated results would make more sense."*

> Of course the model could be calibrated to specific points to better match point data, and this has been done in a number of former model applications (e.g. Forkel et al., 2014). But our objective in this paper is to evaluate how the global model reproduces key variables regionally as well as globally, without further tuning or more specific input data. We thus intended to show that the model results are useful independent of spatial scale and that it can principally be forced with different inputs at different resolutions. Thanks to the reviewer for pointing to the importance of local-scale climate input to reduce error propagation. In response to this comment, we now provide additional evaluations for 6 eddy flux tower locations, the simulations are made with global climate data and with observed meteorological data provided for these locations (see supplementary informations L.: 6-14 and Fig.: S17 (NEE), S18 (Evapotranspiration)) . The model performance improves at 2 stations, but also worsens at 1 station when using the site-specific meteorological data.

*"Technical corrections and some minor comments:*

*1. Please introduce each abbreviation in the manuscript text after it is first used. For example, FAPAR was firstly used in Line 29, not Line 106; GPP firstly in Line 117, not in 163;NEE in Line 108 is better after net ecosystem exchange and could avoid in Line 181. Please also check other abbreviations*

> Thanks for making us aware of this. We have carefully read the entire text regarding the definition of abbreviations and have corrected it respectively.

*"2. FAOstat in Line 110 may be better for FAOSTAT;"*

> We have changed that accordingly.

*"3. In Line 189 "a empirical model" should be an empirical model;"*

> Thanks, done.

*"4. The citation for HWSD data (Line 189) is better for: FAO/IIASA/ISRIC/ISSCAS/JRC, 2012. Harmonized World Soil Database (version 1.2). FAO, Rome, Italy and IIASA, Laxenburg, Austria; "*

> We have added this reference.

*"5. There is double "a" in Line 201, double "in" in Line 313, delete one;"*

> Thanks, done.

*"6. CALM and IPA are firstly used in Line 219; GFED and CCI in 226;"*

> Thanks, we have defined all at their first occurrence.

*"7. A period should be at the end of Line 309;"*

> The period is given by the database and differ for each stand. We use the respective simulation period for the evaluation. We have added a note to explain this in the text (L.: 333-334).

*"8. "soil organic carbon" in Line 188 can be short for SOC;"*

> We have abbreviated this term.

*"9. Soil carbon pool can also compare to (Tian et al., 2015);"*

> Yes, of course this present benchmarking could be extended by some additional datasets. Here we provide a first comprehensive attempt of an extensive benchmarking for key features of the LPJmL4 model. As we will publish the model code, we hope that also the benchmarking will be developed further (also by other groups). Here we use only publicly available data that the benchmarking can be assessed by others as well. But thanks for pointing to the Tian et al. 2015 publication, we will keep this in mind for future studies.

*"10. Line 391 SI-Fig.86 might be SI-Fig.66a for biomass? Also check other SI-Figs."*

Thanks, yes you are right. We have checked it throughout the manuscript.

*"Add linear regression coefficients of slope, intercept, R square (R2), P value and root mean square error (RMSE) for Fig. 4; For Fig.4c, to provide a side by side sub-plot of GPP from MTE against observed data (in SI) can be beneficial."*

We added the slope, $R^2$, p value, the NME and the NMSE as proposed as out metric for the Figures: 4, 6, S1-S18. Additionally, we have prepared also side-by-side plots for GPP and biomass from MTE data against observed data and MTE against model data. All figures are provided in the supplementary information Fig. S68.

*"12. Line 501 MENA is better separated for ME and NA."*

MENA countries are commonly aggregated into one region; we realise that results may differ within the region and among ME and NA, yet such closer evaluation is beyond our scope here.

*"13. Section 3.5 may be too short. Add more details on permafrost area and active-layer depth dynamics."*

You are right, we have extended the section (L: 546 – L:551).

Forkel, M., Carvalhais, N., Schaphoff, S., v. Bloh, W., Migliavacca, M., Thurner, M., and Thonicke, K.: Identifying environmental controls on vegetation greenness phenology through model–data integration, Biogeosciences, 11, 7025–7050, doi:10.5194/bg-11-7025-2014, http://www.biogeosciences.net/11/7025/2014/, 2014.
> We thank Anonymous Referee 2 for his or her constructive comments that we reply to below. Line numbers refer to the marked-up version of the manuscript.

*"Comments on "LPJmL4 – a dynamic global vegetation model with managed land: Part II – Model evaluation"*
*General comments*

*In this manuscript, the authors presented results of model benchmarking of their newly developed model, LPJmL4. They used many contemporary observational (then independent) data for the benchmarking, spanning a wide range of model aspects such as productivity, hydrology, and agriculture. Through this attempt, they clarified characteristics of LPJmL4 in comparison with other models and previous versions. This benchmarking focused on site to global features and so did not go into details of ecological vegetation dynamics, plant physiology, soil biogeochemistry, and human management. Nevertheless, such benchmarking is an increasingly important task for model intercomparison, and this study is a good attempt. The manuscript is, frankly speaking, quite long, although this is the second part of* the full length of their work. Result description of each examined variable may be shortened to some extent (not mandatory). Overall, as a benchmarking paper, this *manuscript is reasonably organized, and I found no logical fault."*

> Thank you for supporting the idea of a comprehensive evaluation of different model features. Due to the amount of data used here we have focused on key processes covered by data which are freely available. The paper is by necessity quite long, but we hope to keep it comprehensible for readers by making use of the SI which contains the more detailed information as opposed to the key information in the main paper.

*"Specific comments*
*1. Line 40: I agree that benchmarking became more and more important and several standardized systems have been proposed. As an example, I suggest referring the iLAMB (https://www.ilamb.org/) as a representative system."*

> Thanks, we have added the iLAMB project as a reference to the introduction part (L.: 43-44).

*"2. Line 61: Harris (2015) does not appear in References."*

> The reference is included and we have cited as recommended by
> http://catalogue.ceda.ac.uk/uuid/5dca9487dc614711a3a933e44a933ad3

*"3. Line 65: Please give full words for NCEP."*

> Done.

*"4. Line 64: As long as I know, all meteorological forcing variables are available from ERA-interim (or other appropriate dataset). By using the single dataset, you could conduct more comprehensive simulations with higher integrity. Why did you use different datasets?"*

You are right, ERA-interim provides all forcing, but we want to stick to observational data which are independently conducted and thus can be used as they are. A second point is that these data are going back until 1901, which represents nearly pre-industrial climate and thus is important for the spinup-phase and the equilibrium state of the vegetation distribution and the carbon pools. We only replaced cloudiness data with radiation data from ERA-interim as these data can be used directly by the model and we think these data are more reliable than cloudiness. Nevertheless the model can be forced by any datasets and an uncertainty analysis in an additional paper due to this fact could be beneficial. We have conducted some additional experiments with observational point data (see supplementary informations L.: 6-14 and Fig.: S17 (NEE), S18 (Evapotranspiration)) with which we are able to show that the data used here and the point data show no great difference in their results.

*"5. Line 141: This sentence could be removed or merged to other sentences."*

Thanks, we have moved this sentence up to the first part of the description (L.:148).

*"6. Line 208: Please add a reference to the FLUXNET data base."*

Thanks, done.

*"7. Line 231: Just confirmation. You did not use any data of solar-induced chlorophyll fluorescence (SIF) for benchmarking FAPAR and GPP. OK? Because SIF is increasingly used in such benchmarking, I suggest at least referring the use of SIF in your forthcoming study."*

We can use FAPAR and GPP data directly for evaluating the model parameter. We need additional assumptions to link SIF data to the model parameter,  and but we reckon also "the translation of fluorescence data to photosynthesis is not trivial" (Meroni et al., 2009), which additionally makes the interpretation within the model very difficult. Zheng et al. (2017) suggested that the comparison of SIF and GPP show qualitatively the same response to droughts but not quantitatively. These different points would need further efforts and discussions which is not the object of this paper.

*"8. Line 310: "For" to "for""*

Thanks for detecting this, done.

*"9. Line 324: What do you mean for "observed mean" of vegetation distribution?"*

Thanks you are right, as we do not introduce the mean model at all we have deleted this part.

"10. Line 336: Remove "call". OK?"

Done.

*"11. Line 389: SI-Fig.87 should be SI-Fig.66."*

Right, done.

*"12. Line 393: Can you explain why such overestimation occurred in vegetation biomass of Carvalhuis et al. (2014)?"*

This explanation is somewhat speculative as we have indicated with the word "probably". But the model run without land use shows similar estimates for these latitudes, which suggests that the land use could be underestimated there. The map of Liu et al. (2015) shows estimates similar to those from LPJmL4 there. We have added this remark to the paper (L.: 421-422)

*"13. Line 418: Why did not you provide global values of GPP and NPP? You did so for irrigation and biomass burning emission."*

Thanks for pointing this out. We have added the global numbers for GPP and NPP in line 447 resp. 449 and for SOC in line 411 as well for vegetation carbon in line 414.

*"14. Line 435: Figure 6. Please add a title and units for x-axis."*

Thanks, we have added both.

*"15. Line 546: "Pg C p.a." to "Pg C yr-1" "*

We went through the text carefully and made units consistent.

*"16. Line 562: Units and numbers of each color scale are difficult to read."*

Figure lettering enlarged now.

*"17. Line 564: "form" to "from"."*

Thanks, done.

*"18. Line 619: Maybe, "beans" is more popular than "pulses" (if correct)."*

Pulses incorporate more than only beans. In our CFT-definition it includes beans, peas and lentils. So we keep pulses.

M. Meroni, M. Rossini, L. Guanter, L. Alonso, U. Rascher, R. Colombo, J. Moreno, Remote sensing of solar-induced chlorophyll fluorescence: Review of methods and applications, In Remote Sensing of Environment, Volume 113, Issue 10, 2009, Pages 2037-2051, ISSN 0034-4257, https://doi.org/10.1016/j.rse.2009.05.003.

Yiqi Zheng, Nadine Unger, Jovan M. Tadić, Roger Seco, Alex B. Guenther, Michael P. Barkley, Mark J. Potosnak, Lee T. Murray, Anna M. Michalak, Xuemei Qiu, Saewung Kim, Thomas Karl, Lianhong Gu, Stephen G. Pallardy, Drought impacts on photosynthesis, isoprene emission and atmospheric formaldehyde in a mid-latitude forest, In Atmospheric Environment, Volume 167, 2017, Pages 190-201, ISSN 1352-2310, https://doi.org/10.1016/j.atmosenv.2017.08.017.

[revised manuscript text omitted]

Correlation Coefficient

Standard Deviation
Centered RMS Difference

| | | |
|---|---|---|
| ◇ AK–Atqasuk | ◇ Gebesee | ◇ Nonantola |
| ◇ Kaamanen_wetland | ◇ Hainich | ◇ OR–Metolius–intermediate_aged_ponderosa_pine |
| ◇ AK–Ivotuk | ◇ Tharandt–old_spruce | ◇ OR–Metolius–first_young_aged_pine |
| ◇ Sodankyla | ◇ Lonzee | ◇ OR–Metolius–second_young_aged_pine |
| ◇ Flakaliden | ◇ Wetzstein | ◇ NH–Bartlett_Experimental_Forest |
| ◇ Hyytiala | ◇ Vielsalm | ◇ Avignon |
| ◇ Skyttorp | ◇ Bily_Kriz–_Beskidy_Mountains | ◇ Puechabon |
| ◇ Norunda | ◇ Bily_Kriz–_grassland | ◇ San_Rossore |
| ◇ UCI–1998_burn_site | ◇ Hesse_Forest–Sarrebourg | ◇ Aurade |
| ◇ Griffin–Aberfeldy–Scotland | ◇ MT–Fort_Peck | ◇ Lamasquere |
| ◇ UCI–1989_burn_site | ◇ Laegeren | ◇ Lecceto |
| ◇ UCI–1964_burn_site | ◇ Oensingen1_grass | ◇ MA–Harvard_Forest_EMS_Tower |
| ◇ UCI–1930_burn_site | ◇ Oensingen2_crop | ◇ Island_of_Pianosa |
| ◇ East_Saltoun | ◇ Neustift/Stubai | ◇ MA–Little_Prospect_Hill |
| ◇ UCI–1981_burn_site | ◇ WI–Mature_red_pine | ◇ Roccarespampani_1 |
| ◇ Lille_Valby | ◇ Bugacpuszta | ◇ Roccarespampani_2 |
| ◇ Soroe–_LilleBogeskov | ◇ Renon Ritten | ◇ Vall_d'Alinya |
| ◇ Loobos | ◇ WI–Lost_Creek | ◇ Amplero |
| ◇ Horstemeer | ◇ WI–Willow_Creek | ◇ Castelporziano |
| ◇ Cabauw | ◇ La_Mandria | ◇ NE–Mead–rainfed_maize–soybean_rotation_site |
| ◇ Brasschaat | ◇ MI–Univ._of_Mich._Biological_Station | ◇ NE–Mead–irrigated_continuous_maize_site |
| ◇ Mehrstedt | ◇ Zerbolo–Parco_Ticino–_Canarazzo | ◇ NE–Mead–irrigated_maize–soybean_rotation_site |

[revised manuscript text omitted]
. S1 - S16. Here we use the standard input as described by Schaphoff et al. (under Revision, Section 2.1). Furthermore, we evaluate the model performance on eddy flux tower sites by using site specific meteorological input data provided by http://fluxnet.fluxdata.org/data/la-thuile-dataset/ (ORNL DAAC, 2011). Here the long time spin up of 5000 years was made with the input data described in Schaphoff et al. (under Revision), but an additional spin up of 30 years was conducted

10 with the site specific input data followed by the transient run given by the observation period. Comparisons are shown for some illustrative stations for net ecosystem exchange (NEE) in Fig. S17 and for evapotranspiration Fig. S18. Only 2 stations show a slightly better performance of LPJmL4 to NEE measurements and for evapotranspiration only 1 station, the others show a similar correlation as the simulations conducted with global climate input.

15 We use gauging station to evaluate the river discharge as an integrated measure (Fig. S19 - S66). Fig. S68a compares the two evaluation data sets against each other. The comparison of the global data set from (Jung et al., 2011) to the local data (Luyssaert et al., 2007) shows that both data sets are in good agreement. The comparison of LPJmL4 against the global data set of Jung et al. (2011) on the local scale (Fig. S68b show a slightly worse match as the comparison against the local data (see

20 Fig. 4, main text). Fig. S68c compares LPJmL4 against the global data set from (Jung et al., 2011), but excluding outliers with very high GPP. That increases the match to these data to a NMSE of 0.69 and a $R^2$ of 0.51. These comparisons show also that the comparisons on both scales are meaningful and can give a good indication how good the model can reproduce global as well as local biomass estimations by different methods. Fig. S67 and Fig. S69a give a comparison with the global estima-

25 tion of Carvalhais et al. (2014) for soil organic carbon resp. biomass. Additionally we have compared aboveground biomass in Fig. S69b with estimates by Liu et al. (2015). A spatial  comparison of ecosystem respiration is shown in Fig. S70.

Evapotranspiration comparison against MTE data (Jung et al., 2011) is shown in Fig. S71 and Fig. S72 shows a comparison of simulated fractional burnt area against remote sensing observations
30 (GFED4: http://www.globalfiredata.org/ and CCI Fire Version 4.1: http://cci.esa.int/data). Remote sensing data are also used for the evaluation of FAPAR (Fig. S73) and Albedo (Fig. S74).

Sowing dates have been proved to be important to simulate crop variability (Fig. S75 - S83), a comparison with MIRCA sowing dates we show in Fig. S84 - S93.

Table S1 gives an overview of estimates for regional field application efficiencies, showing that
35 LPJmL4 are in a similar range as other estimates.

[Figure]

**Figure S1.** Comparison of NEE fluxes with EDDY-flux measurements.

[Figure]

**Figure S2.** Comparison of NEE fluxes with EDDY-flux measurements.

[Figure]

**Figure S3.** Comparison of NEE fluxes with EDDY-flux measurements.

[Figure]

**Figure S4.** Comparison of NEE fluxes with EDDY-flux measurements.

[Figure]

**Figure S5.** Comparison of NEE fluxes with EDDY-flux measurements.

[Figure]

**Figure S6.** Comparison of NEE fluxes with EDDY-flux measurements.

[Figure]

**Figure S7.** Comparison of NEE fluxes with EDDY-flux measurements.

[Figure]

**Figure S8.** Comparison of Evapotranspiration fluxes with EDDY-flux measurements.

[Figure]

**Figure S9.** Comparison of Evapotranspiration fluxes with EDDY-flux measurements.

[Figure]

**Figure S10.** Comparison of Evapotranspiration fluxes with EDDY-flux measurements.

[Figure]

**Figure S11.** Comparison of Evapotranspiration fluxes with EDDY-flux measurements.

[Figure]

**Figure S12.** Comparison of Evapotranspiration fluxes with EDDY-flux measurements.

[Figure]

**Figure S13.** Comparison of Evapotranspiration fluxes with EDDY-flux measurements.

[Figure]

**Figure S14.** Comparison of Evapotranspiration fluxes with EDDY-flux measurements.

[Figure]

**Figure S15.** Comparison of Evapotranspiration fluxes with EDDY-flux measurements.

[Figure]

**Figure S16.** Comparison of Evapotranspiration fluxes with EDDY-flux measurements.

NEE

—— LPJmL4 fluxes

---- Euroflux and Ameriflux Data

[Figure]

**Figure S17.** Comparison of NEE fluxes with EDDY-flux measurements driven by site specific meteorological data.

Evapotranspiration

—— LPJmL4 fluxes

---- Euroflux and Ameriflux Data

[Figure]

**Figure S18.** Comparison of Evapotranspiration fluxes with EDDY-flux measurements driven by site specific meteorological data.

[Figure]

**Figure S19.** Evaluation of river discharge at gauging stations [1].

[Figure]

**Figure S20.** Evaluation of river discharge at gauging stations [2].

[Figure]

**Figure S21.** Evaluation of river discharge at gauging stations [3].

[Figure]

**Figure S22.** Evaluation of river discharge at gauging stations [4].

[Figure]

**Figure S23.** Evaluation of river discharge at gauging stations [5].

[Figure]

**Figure S24.** Evaluation of river discharge at gauging stations [6].

[Figure]

**Figure S25.** Evaluation of river discharge at gauging stations [7].

[Figure]

**Figure S26.** Evaluation of river discharge at gauging stations [8].

[Figure]

**Figure S27.** Evaluation of river discharge at gauging stations [9].

[Figure]

**Figure S28.** Evaluation of river discharge at gauging stations [10].

[Figure]

**Figure S29.** Evaluation of river discharge at gauging stations [11].

[Figure]

**Figure S30.** Evaluation of river discharge at gauging stations [12].

[Figure]

**Figure S31.** Evaluation of river discharge at gauging stations [13].

[Figure]

**Figure S32.** Evaluation of river discharge at gauging stations [14].

[Figure]

**Figure S33.** Evaluation of river discharge at gauging stations [15].

[Figure]

**Figure S34.** Evaluation of river discharge at gauging stations [16].

[Figure]

**Figure S35.** Evaluation of river discharge at gauging stations [17].

[Figure]

**Figure S36.** Evaluation of river discharge at gauging stations [18].

[Figure]

**Figure S37.** Evaluation of river discharge at gauging stations [19].

[Figure]

**Figure S38.** Evaluation of river discharge at gauging stations [20].

[Figure]

**Figure S39.** Evaluation of river discharge at gauging stations [21].

[Figure]

**Figure S40.** Evaluation of river discharge at gauging stations [22].

[Figure]

**Figure S41.** Evaluation of river discharge at gauging stations [23].

[Figure]

**Figure S42.** Evaluation of river discharge at gauging stations [24].

[Figure]

**Figure S43.** Evaluation of river discharge at gauging stations [25].

[Figure]

**Figure S44.** Evaluation of river discharge at gauging stations [26].

[Figure]

**Figure S45.** Evaluation of river discharge at gauging stations [27].

[Figure]

**Figure S46.** Evaluation of river discharge at gauging stations [28].

[Figure]

**Figure S47.** Evaluation of river discharge at gauging stations [29].

[Figure]

**Figure S48.** Evaluation of river discharge at gauging stations [30].

[Figure]

**Figure S49.** Evaluation of river discharge at gauging stations [31].

[Figure]

**Figure S50.** Evaluation of river discharge at gauging stations [32].

[Figure]

**Figure S51.** Evaluation of river discharge at gauging stations [33].

[Figure]

**Figure S52.** Evaluation of river discharge at gauging stations [34].

[Figure]

**Figure S53.** Evaluation of river discharge at gauging stations [35].

[Figure]

**Figure S54.** Evaluation of river discharge at gauging stations [36].

[Figure]

**Figure S55.** Evaluation of river discharge at gauging stations [37].

[Figure]

**Figure S56.** Evaluation of river discharge at gauging stations [38].

[Figure]

**Figure S57.** Evaluation of river discharge at gauging stations [39].

[Figure]

**Figure S58.** Evaluation of river discharge at gauging stations [40].

[Figure]

**Figure S59.** Evaluation of river discharge at gauging stations [41].

[Figure]

**Figure S60.** Evaluation of river discharge at gauging stations [42].

[Figure]

**Figure S61.** Evaluation of river discharge at gauging stations [43].

[Figure]

**Figure S62.** Evaluation of river discharge at gauging stations [44].

[Figure]

**Figure S63.** Evaluation of river discharge at gauging stations [45].

[Figure]

**Figure S64.** Evaluation of river discharge at gauging stations [46].

[Figure]

**Figure S65.** Evaluation of river discharge at gauging stations [47].

[Figure]

**Figure S66.** Evaluation of river discharge at gauging stations [48].

[Figure]

**Figure S67.** The maps (left side) show the spatial pattern of soil organic carbon [gC m$^{-2}$] distribution from the standard LPJmL4 simulation against data from Carvalhais et al. (2014). The graph on the right side shows the latitudinal pattern of vegetation biomass distribution simulated by the different versions of LPJmL4 against data from Carvalhais et al. (2014).

[Figure]

**Figure S68.** Comparison of GPP from different sources; MTE data (Jung et al., 2011) against plot data (Luyssaert et al., 2007) (a), LPJmL4 against MTE data (Jung et al., 2011) (b), and LPJmL4 against MTE data (Jung et al., 2011) but without the outliers of very high GPP in the MTE data (c).

(a)

[Figure]

(b)

**Figure S69.** (a) The maps (left side) show the spatial pattern of vegetation biomass [gC m$^{-2}$] distribution from the standard LPJmL4 simulation against data from  Jung et al. (2011); Carvalhais et al. (2014) . The graph on the right side shows the latitudinal pattern of vegetation biomass distribution simulated by the different versions of LPJmL4 against data from  Jung et al. (2011); Carvalhais et al. (2014) . (b) Similar as above but for aboveground biomass [gC m$^{-2}$] from Liu et al. (2015).

[Figure]

**Figure S70.** Evaluation of ecosystem respiration [gC m$^{-2}$ a$^{-1}$] comparing LPJml4 with satellite-derived ecosystem respiration (Jägermeyr et al., 2014).

[Figure]

**Figure S71.** The maps (left side) show the spatial pattern of evapotranspiration $[\mathrm{mm\,a^{-1}}]$ distribution from the standard LPJmL4 simulation against the MTE data (Jung et al., 2011). The graph on the right side shows the latitudinal pattern of evapotranspiration distribution simulated by the different versions of LPJmL4 against data from Jung et al. (2011).

[Figure]

**Figure S72.** Observed and simulated estimations of fractional area burnt. Observed estimation both are based on remote sensing data (GFED4: http://www.globalfiredata.org/ and CCI Fire Version 4.1: http://cci.esa.int/data).

[Figure]

**Figure S73.** FAPAR comparison of seasonal dynamic for Koeppen-Geiger classification against 3 different remote sensing products: MODIS FAPAR, GIMMS3g FAPAR, and VGT2 FAPAR.

A map of the Köppen classification can be found here [http://koeppen-geiger.vu-wien.ac.at].

[Figure]

**Figure S74.** Albedo comparison for Koeppen-Geiger classification with MODIS remote sensing data.

A map of the Köppen classification can be found here [http://koeppen-geiger.vu-wien.ac.at].

[Figure]

**Figure S75.** Evaluation of crop variability comparing rice yields computed by LPJml4 with FAO yield data.

[Figure]

**Figure S76.** As Fig. S75 for soy.

[Figure]

**Figure S77.** As Fig. S75 for sugarcane.

[Figure]

**Figure S78.** As Fig. S75 for rapeseed.

[Figure]

**Figure S79.** As Fig. S75 for sunflower.

[Figure]

**Figure S80.** As Fig. S75 for millet.

[Figure]

**Figure S81.** As Fig. S75 for peanut.

[Figure]

**Figure S82.** As Fig. S75 for cassava.

[Figure]

**Figure S83.** As Fig. S75 for sugar beet.

[Figure]

**Figure S84.** Evaluation of sowing dates of rice: (from top to bottom panel) simulated (LPJmL4) sowing date, observed (MIRCA2000) sowing date and difference between simulated and observed sowing date. Green colours (red colours) in the difference map indicate that simulated sowing dates are too late (too early) compared to observations. White colours indicate crop area smaller than 0.001% of grid cell area. Sowing dates in regions without seasonality are not shown.

[Figure]

**Figure S85.** Evaluation of sowing dates of maize: Caption as for Fig.S84.

[Figure]

**Figure S86.** Evaluation of sowing dates of millet: Caption as for Fig.S84.

[Figure]

**Figure S87.** Evaluation of sowing dates of pulses: Caption as for Fig.S84.

[Figure]

**Figure S88.** Evaluation of sowing dates of sugarbeet: Caption as for Fig.S84.

[Figure]

**Figure S89.** Evaluation of sowing dates of cassava: Caption as for Fig.S84.

[Figure]

**Figure S90.** Evaluation of sowing dates of sunflower: Caption as for Fig.S84.

[Figure]

**Figure S91.** Evaluation of sowing dates of soybean: Caption as for Fig.S84.

[Figure]

**Figure S92.** Evaluation of sowing dates of groundnut: Caption as for Fig.S84.

[Figure]

**Figure S93.** Evaluation of sowing dates of rapeseed: Caption as for Fig.S84.

]

[Figure]

**Figure S94.** Simulated sowing dates of rainfed sugar cane.

**Table S1.** Comparison of field application efficiencies

[revised manuscript text omitted]

---

## Author Response (AR2)

Dear Jules,

we appreciate that you accompanied the review process of the manuscript. Here as well, we have carefully reviewed the text again and found some wrong figure captions and other little things.

We have added in the first paragraph of the conclusion some statements about the performance of the LPJmL4 model for the different evaluations and their importance for the overall performance. However, the individual sections also conclude to their respective evaluations.

More than in the companion paper we need to acknowledge data providers and thus we included some inevitable links.

For both papers we have created a doi for the data and LPJmL4 code. Unfortunately, we still have some open legal issues which we hope that it will be solved until the publication. Until then the code will be available via the doi upon request only, later it will be downloadable over this doi and the gitlab server at PIK. The data are already published and can be downloaded directly via the doi.

Yours faithfully,
Sibyll Schaphoff

***Topical Editor Decision: Publish subject to minor revisions (review by editor)*** *(15 Jan 2018) by Julia Hargreaves*
*Comments to the Author:*
*In the conclusion it is stated that the model results are adequate. My question is, adequate for what? We all know that no model is perfect, and indeed that no observations are perfect, and that in some cases there may be representation error making models and observations difficult to compare. The question is really what does the level of performance of the model say about what the problems the model may be successfully used to tackle (and of course the opposite - in what ways is it not useful). I think that a few comments on this topic may improve the usefulness of this paper!*

*Please make the code available, then describe how it may be obtained. Please also provide the DOI for the exact version of the code described in the paper.*

*Other materials may be uploaded to a public repository with a DOI, or added to the supplement. Weblinks are very ephemeral. Try to include as few weblinks as possible in the final version of the paper.*

*This paper and its companion paper are fairly epic. They could really do with clickable tables of contents, but unfortunately that is not supported within the journal formal. I am very grateful to the 4 reviewers for their carefully considered reviews.*

*For such long papers in GMD it is important that as many of the details are correct as possible. Readers may well dip into the papers, and make use of small details. If there are errors then this considerably reduces the utility of the paper. At the same time, it is unlikely that any of the peer review team are in a position to find these kind of errors. So, you can either check the paper very carefully yourselves at this stage, or you could also ask people who know the model well but are not authors on the paper to check the content for errors. It is up to you, but the latter method has been employed with considerable success (i.e. some important errors were found!) for other long papers in GMD.*

[revised manuscript text omitted]

---

## Author Response (AR3)

Dear Jules,

we appreciate that you accompanied the review process of the manuscript. We have revised the "Code and data availability" section. And we also added the difference file from the last version.

We have added in the first paragraph of the conclusion some statements about the performance of the LPJmL4 model for the different evaluations and their importance for the overall performance. However, the individual sections also conclude to their respective evaluations.

More than in the companion paper we need to acknowledge data providers and thus we included some inevitable links.

For both papers we have created a doi for the data and LPJmL4 code. Data are available under the doi: http://doi.org/10.5880/pik.2017.009 and the code can be downloaded from the PIK's gitlab server: https://gitlab.pikpotsdam.de/lpjml/LPJmL and is available under the doi: http://doi.org/10.5880/pik.2018.002.

Yours faithfully,
Sibyll Schaphoff

***Topical Editor Decision: Publish subject to minor revisions (review by editor)*** *(15 Jan 2018) by Julia Hargreaves*
*Comments to the Author:*
*In the conclusion it is stated that the model results are adequate. My question is, adequate for what? We all know that no model is perfect, and indeed that no observations are perfect, and that in some cases there may be representation error making models and observations difficult to compare. The question is really what does the level of performance of the model say about what the problems the model may be successfully used to tackle (and of course the opposite - in what ways is it not useful). I think that a few comments on this topic may improve the usefulness of this paper!*

*Please make the code available, then describe how it may be obtained. Please also provide the DOI for the exact version of the code described in the paper.*

*Other materials may be uploaded to a public repository with a DOI, or added to the supplement. Weblinks are very ephemeral. Try to include as few weblinks as possible in the final version of the paper.*

*This paper and its companion paper are fairly epic. They could really do with clickable tables of contents, but unfortunately that is not supported within the journal formal. I am very grateful to the 4 reviewers for their carefully considered reviews.*

*For such long papers in GMD it is important that as many of the details are correct as possible. Readers may well dip into the papers, and make use of small details. If there are errors then this considerably reduces the utility of the paper. At the same time, it is unlikely that any of the peer review team are in a position to find these kind of errors. So, you can either check the paper very carefully yourselves at this stage, or you could also ask people who know the model well but are not authors on the paper to check the content for errors. It is up to you, but the latter method has been employed with considerable success (i.e. some important errors were found!) for other long papers in GMD.*

[revised manuscript text omitted]

Correlation Coefficient

Standard Deviation
Centered RMS Difference

Legend (sites, colours, ordered north to south):

- Kaamanen_wetland
- AK–Ivotuk
- Sodankyla
- Flakaliden
- Hyytiala
- Norunda
- Griffin–Aberfeldy–Scotland
- UCI–1930_burn_site
- UCI–1964_burn_site
- East_Saltoun
- UCI–1981_burn_site
- Lille_Valby
- Soroe–LilleBogeskov
- Loobos
- Horstermeer
- Cabauw
- Brasschaat
- Mehrstedt

- Gebesee
- Hainich
- Tharandt–old_spruce
- Lonzee
- Wetzstein
- Vielsalm
- Bily_Kriz–_Beskidy_Mountains
- Bily_Kriz–_grassland
- Hesse_Forest–Sarrebourg
- Neustift/Stubai
- WI–Mature_red_pine
- Bugacpuszta
- Renon/Ritten
- WI–Lost_Creek
- WI–Willow_Creek
- La_Mandria
- MI–Univ._of_Mich._Biological_Station
- Zerbolo–Parco_Ticino–_Canarazzo

- Le_Bray
- Nonantola
- OR–Metolius–old_aged_ponderosa_pine
- OR–Metolius–first_young_aged_pine
- OR–Metolius–intermediate_aged_ponderosa_pine
- OR–Metolius–second_young_aged_pine
- NH–Bartlett_Experimental_Forest
- Avignon
- Puechabon
- San_Rossore
- Aurade
- Lamasquere
- Lecceto
- MA–Little_Prospect_Hill
- MA–Harvard_Forest_EMS_Tower
- Roccarespampani_1
- Roccarespampani_2
- Vall_d'Alinya

- Amplero
- Castelporziano
- Borgo_Cioffi

[revised manuscript text omitted]